# Environmental connectivity controls diversity in soil microbial communities

Manupriyam Dubey [1], Noushin Hadadi [1], Serge Pelet [1], Nicolas Carraro [1], David R. Johnson [2] & Jan R. van der Meer [1]✉

Interspecific interactions are thought to govern the stability and functioning of microbial communities, but the influence of the spatial environment and its structural connectivity on the potential of such interactions to unfold remain largely unknown. Here we studied the effects on community growth and microbial diversity as a function of environmental connectivity, where we define environmental connectivity as the degree of habitat fragmentation preventing microbial cells from living together. We quantitatively compared growth of a naturally-derived high microbial diversity community from soil in a completely mixed liquid suspension (high connectivity) to growth in a massively fragmented and poorly connected environment (low connectivity). The low connectivity environment consisted of homogenously-sized miniature agarose beads containing random single or paired founder cells. We found that overall community growth was the same in both environments, but the low connectivity environment dramatically reduced global community-level diversity compared to the high connectivity environment. Experimental observations were supported by community growth modeling. The model predicts a loss of diversity in the low connectivity environment as a result of negative interspecific interactions becoming more dominant at small founder species numbers. Counterintuitively for the low connectivity environment, growth of isolated single genotypes was less productive than that of random founder genotype cell pairs, suggesting that the community as a whole profited from emerging positive interspecific interactions. Our work demonstrates the importance of environmental connectivity for growth of natural soil microbial communities, which aids future efforts to intervene in or restore community composition to achieve engineering and biotechnological objectives.

[1] Department of Fundamental Microbiology, University of Lausanne, Lausanne, Switzerland. [2] Department of Environmental Microbiology, Swiss Federal Institute of Aquatic Science and Technology, Eawag, Dübendorf, Switzerland. ✉email: janroelof.vandermeer@unil.ch

Natural "free-living" microbial communities and those in association with hosts are exemplified by complex and high-density interspecific interactions, being composed of dozens (e.g., certain insect hosts)[1] up to many thousands (e.g., soils[2]) of individual genotypes that live within short distances from each other in suspension[3] or on surfaces (μm to mm scale)[4,5]. Understanding the general principles governing the formation, structure and functioning of microbial communities is one of the major challenges in microbial ecology, yet our understanding remains largely fragmentary[6–9]. It is generally assumed that interspecific interactions shape the community's functioning and stability within the available nutrient context[10,11], and physico-chemical boundary conditions of the system or the host[12–14]. However, one would expect that the extent to which interspecific interactions can unfold is dependent on the spatial structure of the microbial living environment[15], the distance between cells[16] and its degree of connectivity (Fig. 1a). We postulate here that environmental connectivity is an important yet poorly understood driver for microbial diversity.

To understand how environmental connectivity may influence microbial growth and diversity, consider, for example, an isolated environment that can physically hold only a few cells, such as a water-filled capillary in the soil (Fig. 1a). Because of its physical isolation, there is low or zero connectivity in terms of cell movement or nutrient exchange[13], and the few occurring cells may multiply depending on the local substrate availability (Fig. 1a). If the cells belong to a single genotype, any growth will be solely determined by their inherent physiological capacities. However, as soon as two different genotypes are present, interspecific interactions begin to play a role[16], potentially influencing the growth of each partner (both positively or negatively). The more genotypes that are contained, the higher the number of potential interspecific interactions, but the less likely that any single interaction may dominate the formation and functioning of the microcommunity of cells (Fig. 1a). The reason for this may lie in non-transitiveness, such as the rock-paper-scissors effect (i.e., each genotype "dominates", at least to some extent, another genotype), as was shown recently in a synthetic bacterial species mixture[17].

The connectivity of any environment is thus expected to impose a range of challenges to, limitations on and selection pressures on microbial growth. A community in an environment disconnected into small isolated spaces, where each space can only hold a few cells and genotypes, will be restricted to isolated growth dominated by lower order interspecific interactions. The same community in a connected spatial environment can develop higher order interspecific interactions because more genotypes are likely to be in contact, but their individual effects are subdued. Consequently, environmental connectivity should affect microbiome structure and microbial diversity, but this has yet to be systematically studied. Our main aim, therefore, was to understand how and to what extent environmental connectivity influences the growth and diversity of soil microbial communities. Our hypothesis, based on the reasoning above, was that low connected environments would be unfavorable for community growth and diversity. This would occur if, on average, the founder genotypes are few and predominantly negatively affect each other. In comparison, growth environments with high connectivity would be favorable for the proliferation of many species simultaneously as a consequence of non-transitive dynamics.

In order to study the effect of environmental connectivity, we designed an experimental system to follow and quantitatively compare the growth of a highly diverse microbial community derived from a natural soil in high and low connectivity environments. For the high connectivity environment, we incubated the starting community in a completely mixed and suspended environment. For the low connectivity environment, we separated individuals from the same community into different homogenously sized agarose microbeads, where each bead initially contained 1–2 founder cells and beads were incubated *en masse*. Embedding cells in agarose microbeads restricts cell movement but not substrate diffusion. We further expected that interspecific interactions mediated by e.g., diffusible molecules primarily affect paired species at short distances, and thus occurs within but not between individual beads[16]. We measured global growth rates and deduced diversity changes from 16S rRNA gene amplicon sequencing. We quantified individual and paired cell growth in beads from microcolony size estimations using time series microscope imaging, which were further used to deduce the magnitude (e.g., strong or weak) and global nature (e.g., positive or negative) of interspecific interactions within the community. To gain more insight into the role of environmental connectivity on community productivity and diversity, we developed a computational model that simulates the growth of individual genotypes in communities for high and low connectivity environments using Monod-type substrate kinetics, while accounting for the initial amount of viable cells and globally attributed interspecific interactions. Our results, supported by community growth simulations, indicate that a pool of fragmented and homogenously sized low connectivity environments disfavors maintenance of microbial diversity compared to high connectivity environments. Despite observing the expected increased dominance of negative interspecific interactions, randomized partnerships of soil bacteria were beneficial for growth of the community across all beads when compared to growth of single founder cells inside individual beads. This suggests that soil communities may have evolved to profit from random partnerships in fragmented and highly structured environments.

## Results

**Experimental design of low and high environmental connectivity environments**. To experimentally quantify the effects of environmental connectivity on growth and diversity, we designed a system to produce multitudes of individual low-connectivity environments. Each individual environment contains 1–2 founder cells, while the system as a whole (i.e., across all the individual low connectivity environments) retains high microbial diversity. We compared low connectivity environment to a high-connectivity environment that contains the same overall high microbial diversity and number of founder cells (Fig. 1). We used soil to obtain a microbial community with high diversity, as soil microbial communities contain among the highest levels of microbial species richness on a per gram basis of all microbiomes[2,18,19]. To avoid culturing bias, we washed the cells freshly from the same soil for each experiment, and used this starter community directly for the various incubations (*sand community* or SC, Fig. 1b). To produce a low connectivity environment, we randomly encapsulated SC cells inside agarose beads with 40–70 μm diameter (Fig. 1c). This spatially isolates 1–2 individal founder cells into an individual agarose bead and creates a local environment, and promotes the development of interspecific interactions[16] (Fig. 1c and Supplementary Table 1). To produce a high connectivity environment, we propagated the community in well mixed and suspended batch culture (Fig. 1d), which eliminates long-term close spatial contact of genotypes and represses the development of interspecific interactions. Both liquid (high connectivity) and agarose beads (low connectivity) contained the same nutrient medium and, at the system-level, the same starting SC genotypes and cell numbers.

**Environmental connectivity has no effect on aggregate community growth**. Since we had no a priori knowledge on the

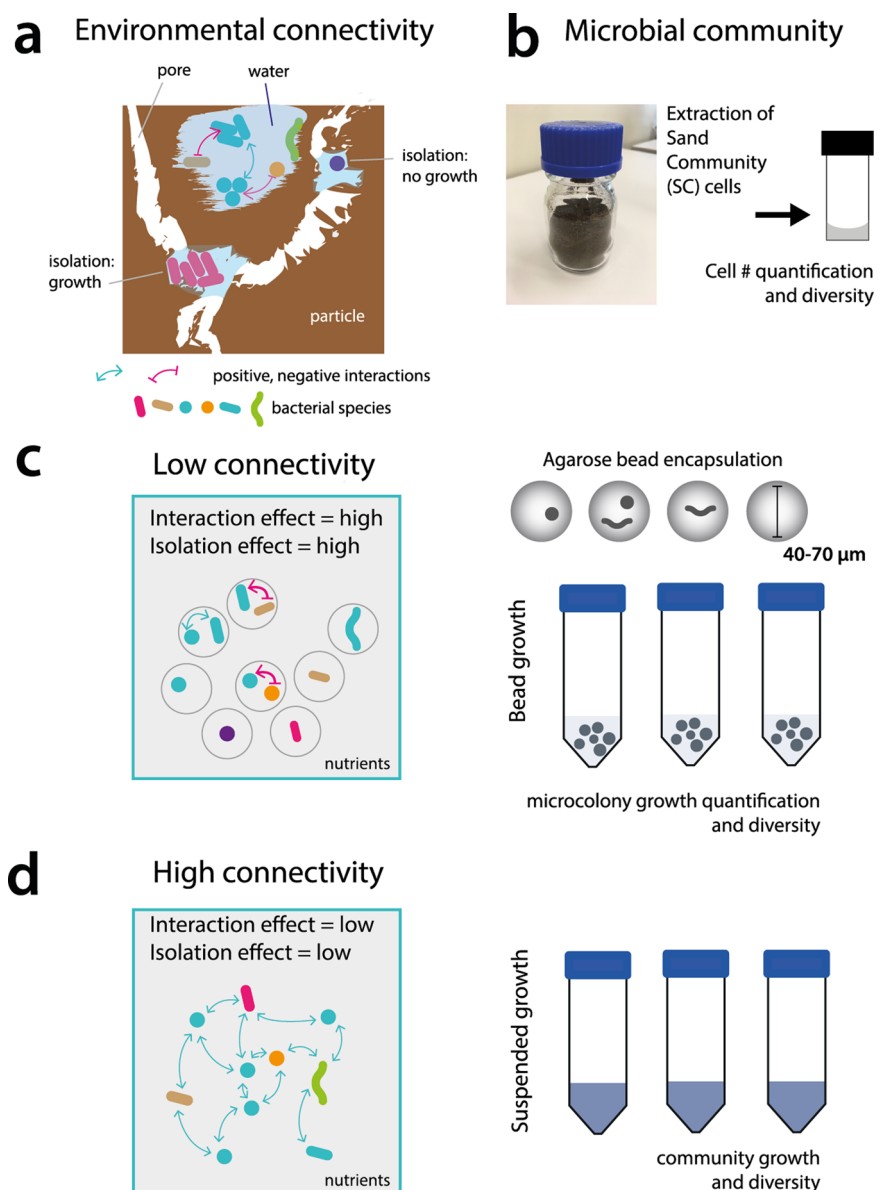

**Fig. 1 Environmental connectivity and its experimental design. a** Soil microbial living environments are characterized by spatially fragmented and isolated pockets, such as formed by soil particles and water filled pores. Different species numbers may assemble depending on the volume of disconnected space, which is expected to directly influence the resulting growth properties and the extent of developing interspecific interactions of the (sub)communities. **b** A highly diverse microbial community used as inoculant is freshly extracted and dispersed from sand (SC) and quantified. **c** Growth under low connectivity, consisting of millions of disconnected environments formed by randomly encapsulating 1–2 founding SC cells in small substrate-permeable agarose beads, incubated in parallel. Cells under low connectivity are expected to face a strong isolation effect when being alone and strong interaction terms (red arrows, inhibition; blue arrows, neutral) when starting off in random partnerships. **d** Growth under high connectivity in a mixed liquid suspension of the same starting SC cell numbers and diversity as in **c**. High connectivity is expected to yield low isolation and low interspecific interaction effects, as a result of random mixing.

growth properties of freshly prepared SC cells, we examined various aspects of their growth in high (liquid culture) and low (beads) connectivity environments. We compared SC growth to that of a pure culture of the soil bacterium *Pseudomonas veronii* strain 1YdBTEX2[20,21] under the same conditions to account for potential effects of the bead-production procedure. Between 76.4 and 84.4% of cells in the washed SC cell suspension before encapsulation stained positively with propidium iodide as assessed by flow cytometry. This suggests that most of them were physiologically impaired. Despite this, however, SC cell numbers increased in mixed liquid suspension at rates only slightly slower than *P. veronii* in pure culture (Fig. 2a, b). SC growth in beads

was evident from increased SC microcolony sizes ($t = 6$, 24, or 48 h, Fig. 2a), with some cell dispersion within the beads at later time points ($t = 48$ or 72 h, Supplementary Table 1). Using the smallest and weakest stained fluorescent pixel objects imaged within beads as a threshold, we estimated that 20–30% of the encapsulated SC cells and 6–15% of the *P. veronii* cells may not have divided within the beads (Fig. 2c). This indicated that the majority of SYTO-9 detectable SC cells in the initial preparation were viable and capable of dividing. Propidium iodide-positive cells in the washed starting SC suspension stain poorly with SYTO-9 and, consequently, may not have been further detected. Furthermore, this showed that the bead process itself is not

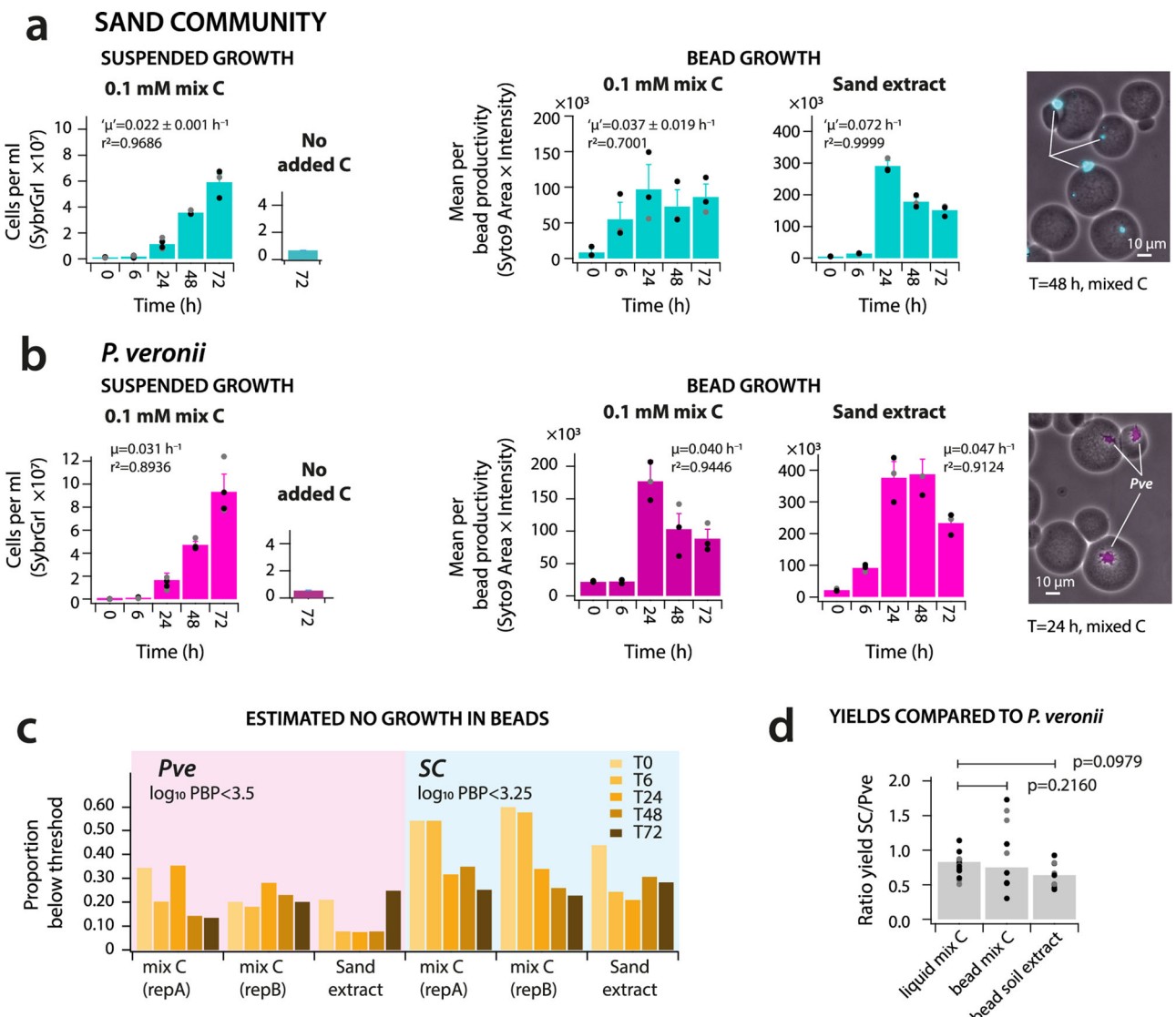

**Fig. 2 Aggregate community growth is unaffected by environmental connectivity. a** Growth of SC cells under high (i.e., mixed liquid suspension, in cells ml$^{-1}$) and low connectivity conditions (i.e., encapsulated in beads). Growth in beads is expressed as the mean per bead productivity (PBP). PBP is defined as the product of imaged microcolony areas times their mean SYTO-9 fluorescence intensity summed per bead, averaged across all imaged beads ($n = 100-500$ beads) in a replicate series. Mix C, mixture of 16 different carbon substrates (equimolar, to 1 mM final carbon concentration); sand extract, carbon and nutrient solution extracted from sand. Bars show mean community cell numbers of four (liquid) or three (beads) biological replicates, ± one *SD*, with individual data points. "μ" derived maximum community growth rate. Image shows detail of microcolonies (in cyan) in beads after 48 h. **b** as **a** but for a pure culture of the soil bacterium *P. veronii*. **c** Estimated proportion of non-growing *P. veronii* (*Pve*) or SC cells in bead incubations below the defined PBP thresholds. **d** SC community yield comparison across high and low connectivity conditions, taken as the ratio of SC yield to that of *P. veronii* under the same conditions. *p*-values from *t*-test (two-sided, unequal variance) of individual ratios, $n = 12$ (liquid) or 9 (beads). Bars show mean ratios with symbols presenting individual data points.

damaging or inhibiting to the division of the majority of cells. The increase in the maximum observed SC microcolony size after 24–72 h compared to $t = 0$ translates to approximatetly 9–10 generations of cell division per bead under the imposed carbon substrate regime (assuming a round packed colony of cells, Supplementary Note 1).

The estimated aggregate community growth rates were comparable for suspended and bead growth when using the same defined mixed carbon substrate (Fig. 2a; two-sided *t*-test, $p = 0.0952$, $n = 3$). SC growth in beads was faster with sand extract than with mixed C substrate (Fig. 2a; two-sided *t*-test, $p = 0.0281$, $n = 3$), suggesting that the diversity of substrates extracted from sand is favorable for community-level proliferation. Because of the different methodologies to quantify biomass in the high

connectivity (cell counting) and low connectivity (microcolony imaging) environments, we quantified SC yields for the different environments relative to those for *P. veronii* under the same conditions (Fig. 2d). We further quantified SC yields by quantifying the mass of isolated DNA from sub-samples. The ratios of SC to *P. veronii* yields were not significantly different between liquid suspended and bead growth (Fig. 2d; two-sided *t*-tests, $p > 0.1$). Isolated DNA concentrations from SC in low or high connectivity were also not significantly different ($p = 0.10$, Supplementary Table 2), indicating that the aggregate community yields of SC in high and low connectivity environments were similar. These experiments thus indicated that the sand-derived microbial community as a whole was capable of growing under both experimental conditions, and suggested that its aggregate

growth properties (rate and yield) were not affected by the degree of connectivity of the growth environment.

**High environmental connectivity favors maintenance of higher species diversity**. Although the aggregate growth properties of SC cells were comparable for low and high connectivity environments, this did not take into account any relative changes in genotype abundances. The starting composition of washed SC cells deduced from 16S rRNA gene amplicon sequencing had a taxonomic richness of approximately 1200 operational taxonomic units (OTUs, summed from independent washed SC replicates; mean richness per replicate $= 543 \pm 153$, $n = 7$), covering all known soil bacterial groups (Fig. 3a). SC growth in low connectivity environments dramatically reduced OTU richness by more than ten-fold after 48 h, irrespective of the growth substrate and with near extinction (i.e., not detected by sequencing) of several major phyla (Fig. 3a, range of mean richness = 34.5–57, $p$-value = 0.00011, two-sided $t$-test). The diversity decrease was not due to the encapsulation procedure, as higher diversity was still apparent after 6 h in the low connectivity environment (Fig. 3a, Bead growth $t = 6$ h). Growth in the low connectivity environment also skewed the distribution of OTU relative abundances (Fig. 3b). After 48 h, one-third of the observed taxa had increased their relative abundance by a factor of two, while half had decreased relative abundance by a factor of two when compared to the SC composition before encapsulation (Supplementary Fig. 1). SC communities prepared on different occasions were clearly different but converged over incubation time into distinguishable high and low connectivity environment trajectories (Fig. 3c). Community diversity remained significantly higher in suspended culture (high connectivity) than in beads (low connectivity) (Fig. 3d, four different diversity measures), although diversity also significantly reduced after 72 h of growth compared to the starting SC composition (Fig. 3a; mean richness = 159 ± 43; two-sided $t$-test $p = 0.000309$, $n = 4$). SC growth in the high connectivity environment retained all major soil phyla (Fig. 3a), with a more similar abundance distribution compared to $t = 0$ (Fig. 3b) and nearly equal proportions of species increasing and decreasing their relative abundances (Supplementary Fig. 1). Collectively, these experiments showed that growth in the high connectivity environment (mixed suspended culturing) is more favorable for maintaining OTU diversity than in the low connectivity environment (cells in isolation in beads).

**Isolated growth in the low connectivity environment is penalized**. In order to understand, why SC growth in the low connectivity environment caused a drastic reduction in OTU diversity despite the cells having access to the same carbon substrates, we analyzed microcolony growth in beads more closely. We first compared the growth of SC cells that were individually enclosed within beads (i.e., *single* occupancy, or that may have had a dead undetectable partner) to that of encapsulated *pairs* of founder cells. Across six independently started encapsulation and incubation experiments, and on two different substrate regimes (i.e., mixed C substrates and sand extract), per bead productivity (PBP) of pairs was significantly higher by more than 2-fold than that for singly occupancy in beads. This was true even though all beads were residing in the same growth medium and culture conditions (Fig. 4a, b and Supplementary Fig. 2; Wilcoxon rank-sum test of median and 75th percentile PBP comparisons at all time points except $t = 0$, $p$-value range 0.00195–0.01367). In addition, only 19.7% (95% confidence interval 14.1–25.3) of beads with single occupancy surpassed a 10-fold productivity increase compared to the start, versus 47.4% (37.5–57.3, $n = 12$) for beads with two or more microcolonies. This indicated that being

randomly partnered was on average beneficial for productivity (i.e., normalized per bead, individual low connectivity environment), whereas being in isolation on average led to comparatively poorer growth. The isolation in the low connectivity environment may therefore have penalized growth of the majority of SC taxa, leading to their near-extinction (below sequencing threshold) after 9–10 generations at the benefit of some 20% opportunistic growers and those in pairs.

To verify this tendency of partner benefit further, we artificially mixed a non-auxotrophic pure *P. veronii* culture (constitutively expressing an mCherry fluorescent protein for discrimination from SC cells) with the SC cell suspension in a 1:1 ratio, encapsulated the cells, and compared growth properties with those of either SC or *P. veronii* alone (Supplementary Fig. 3). Also here, PBP distributions of SC cells paired with *P. veronii* within the same bead largely shifted to higher values compared to those SC growing in single occupancy beads in the same flask (Supplementary Fig. 3; Fisher's test on PBP distributions, $p$-value range 0.0005–0.002). This experiment thus indicated that random SC cells also profited from being paired with an external arbitrary partner pure culture, suggesting general benefits or maybe even consistent needs for paired interactions.

**Despite benefits at the community level, most paired interactions result in imbalanced growth**. Although beads with two or more SC microcolonies showed higher productivities at the system level (i.e., across all beads) than expected from combined single occupancy growth, this per se did not reveal any interaction signs (i.e., positive, neutral, or negative). To look closer at prevalent interspecific interactions, we selected those beads with exactly two partners (Fig. 4c). Paired SC-interactions on sand extract showed 100-fold larger microcolony sizes than on mixed C substrates (e.g., Fig. 4c, 24 h), reflecting the faster and more abundant growth on that substrate (Fig. 2a). When categorizing productivities according to the above-defined imaging threshold of "no-growth" (i.e., $\log_{10}$ PBP < 3.25, Fig. 2c), it became clear that paired interactions were highly non-random (Wilcoxon rank test, $p = 0.0235$ for sand extract and $p = 7.46 \times 10^{-4}$ for mixed C, Fig. 4d). More than half of the observed pairs showed highly imbalanced growth (Fig. 4d, red and magenta zones), whereas under this categorization definition an estimated 19–38% of pairs showed balanced growth (Fig. 4d, blue, within tenfold PBP ratio of partners). Collectively, these observations demonstrated that cells from a highly taxa-rich microbial community from soil, randomly placed in a multitude of parallel low connectivity environments are on average growth disfavored when being alone and favored when being partnered. The majority of paired growth ratios is more than tenfold deviating from one (i.e., equal growth in pairs), which might be the result of interspecific interactions or of differences in inherent growth rates (see below).

**A model to simulate community growth for different connectivity environments and interspecific interactions**. To further substantiate the experimental observations and, in particular, to better understand the potential effects of interspecific interactions on paired growth, we developed a model to simulate community growth and genotype diversity in high or low connectivity environments. The simulation was initiated with a randomly seeded community of $2 \times 10^5$ in silico cells with richness and relative OTU abundances equaling those in the measured SC samples, including or excluding a proportion of dead or growth-compromised cells (e.g., Fig. 3a, see "Methods" section, Supplementary Methods, Section 1). The starting community is then grown in silico with the growth rates observed in our experiments (e.g., Fig. 2a). The growth of each OTU population is

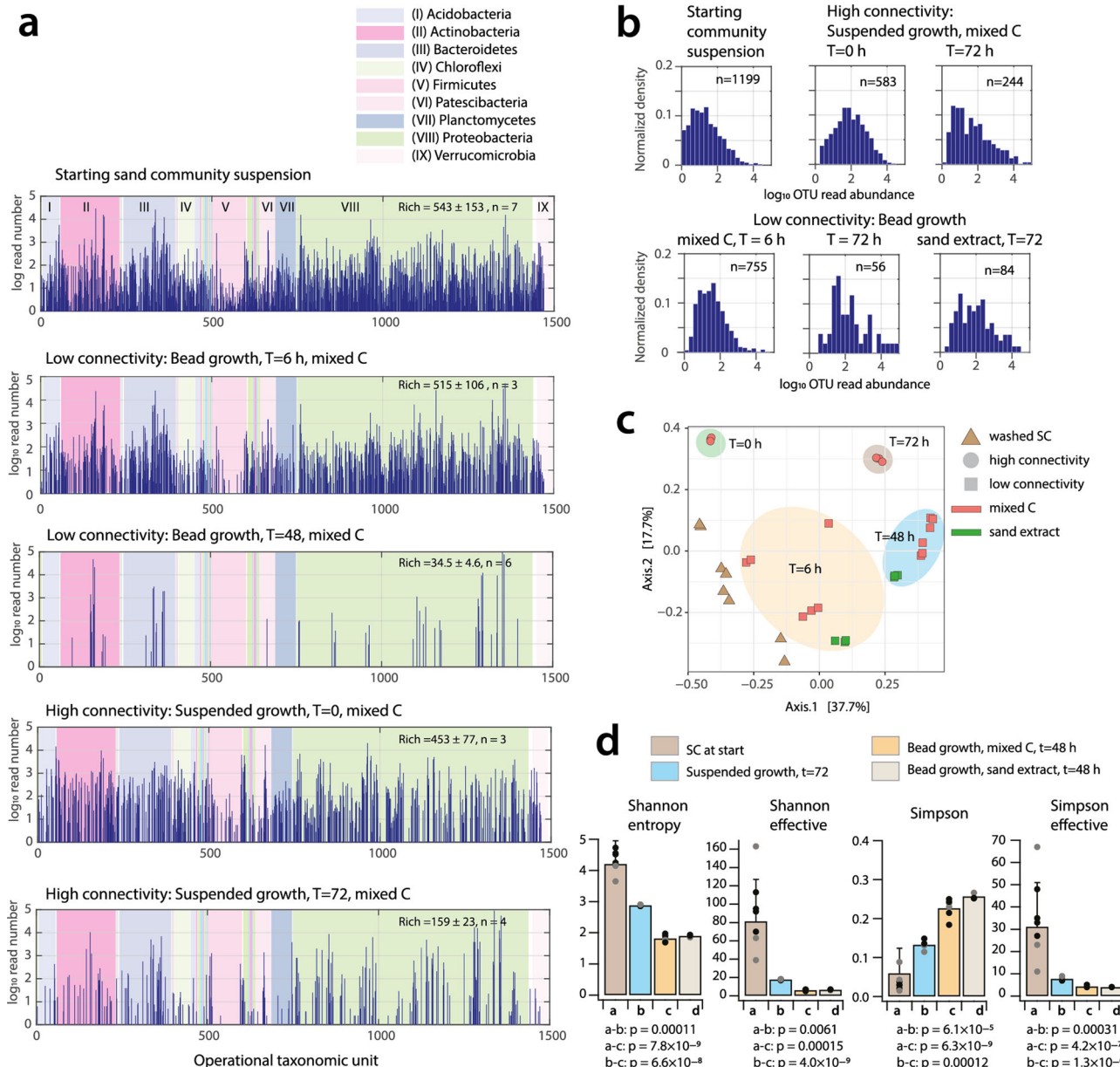

**Fig. 3 Community diversity decrease under low environmental connectivity. a** OTU losses upon growth incubation under low or high connectivity conditions. Log$_{10}$ OTU read abundances of summed replicates per treatment or sample, ranked according to the SILVA-genus-level taxonomic classification (0–1500). Major bacterial phyla indicated in different background colors and Roman numbering. Rich, mean sample richness ± one *SD*, with *n* indicating the number of replicates. **b** Normalized distribution of log$_{10}$ OTU summed read abundances, with *n* indicating the number of OTUs. **c** Multidimensional scaling analysis based on Bray–Curtis distance of SC cell suspensions at start, low (i.e., beads) and high (i.e., suspended growth) connectivity samples, as a function of incubation time and substrate. Percentages indicate explained variation among data sets and replicates. Colored zones manually added to group related samples. **d** Alpha diversity measures for community diversity under high or low connectivity conditions compared to SC at start. Bars show means ± 1 SD plus individual data points, calculated for the replicate sets of panel **a**. Values below are *p*-values from two-sided *t*-test with unequal variance for the indicated comparisons. *p*-Values between both bead regimes are not reported and show no statistically significant difference.

computed in discrete time steps as a function of substrate utilization, interspecific interaction terms, or single-growth penalties using Monod kinetics (see "Methods" section).

For simulations in the absence of empirical data on individual OTU growth rates of SC cells, we hypothesized two growth rate distributions and examined which of the two would be more likely to result in the observed community distribution at stationary phase. In the first distribution, maximum specific growth rates ($\mu_{\mathrm{max,sp}}$) drawn randomly between 0.01 and 0.6 h$^{-1}$ were assigned to OTUs, whereas in the second they were attributed to OTUs

according to a probability function reflecting their measured relative abundance at $t = 0$ h (Supplementary Methods, Section 2). For these two simulations, we used liquid suspended growth (high connectivity environment), such that growth rates are not affected by interspecific interactions (Supplementary Methods, Section 2). Growth simulations with both $\mu_{\mathrm{max,sp}}$ distribution assumptions resulted in distributions of steady-state taxa abundances similar to the experimentally observed ones (Fig. 5a, $p = 0.2649$ two-sided *t*-test, $n = 5$ simulations). However, the distribution where growth rates were biased by the original OTU proportion generated better

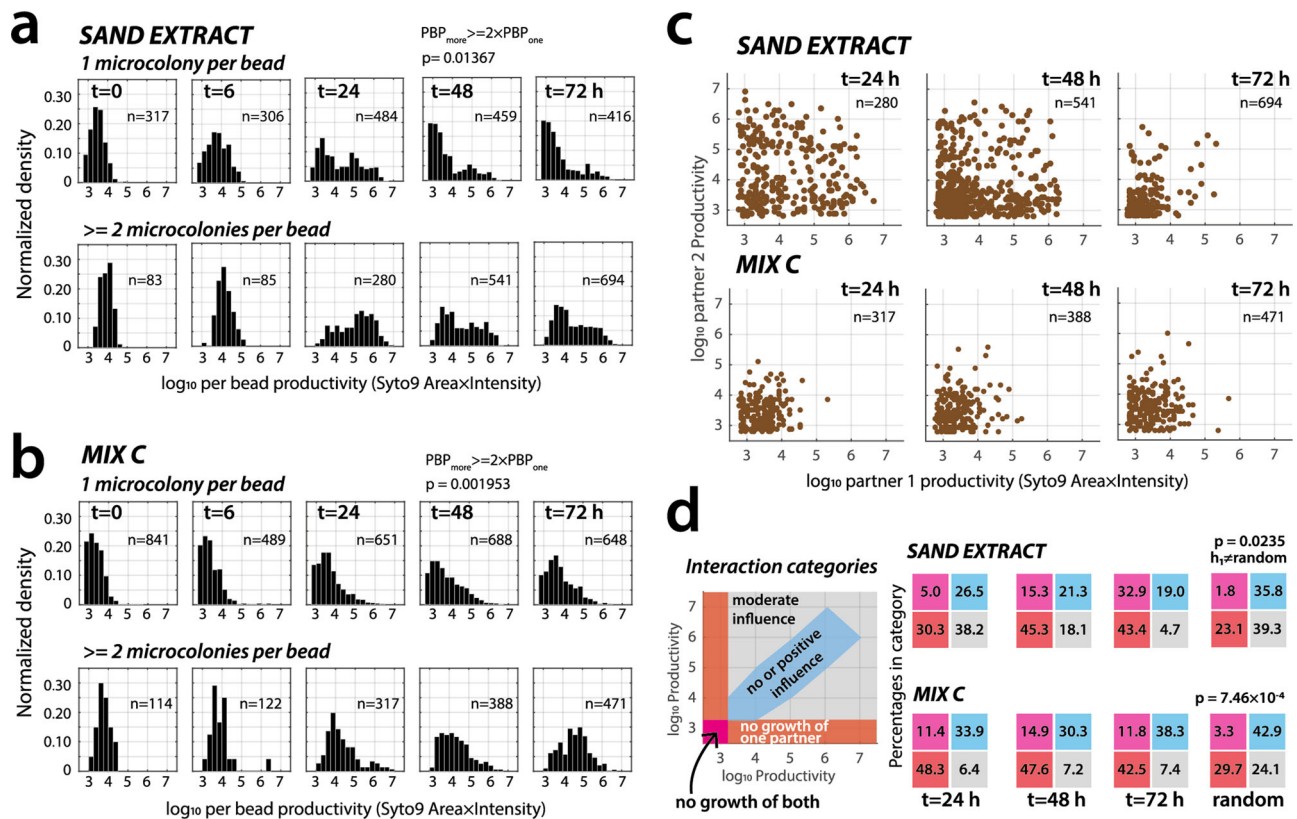

**Fig. 4 Paired growth is favorable over single occupancy growth but globally dominated by negative interactions.** Normalized $\log_{10}$ PBP distributions over time of beads with single occupancy or with two and more microcolonies (mean range: 2.1–3.8) for SC incubated with sand extract (**a**) or with mixed C substrates (**b**). *p*-values are the likelihood that the 75th percentile of all PBP distributions for single occupancy is equal to those of the PBP-values in multiple occupancy, corrected by their mean number of per bead microcolonies at that sampling time point (Wilcoxon rank sum test). *n* = number of beads. **c** Paired PBPs of arbitrarily ranked SC microcolonies inside single beads, presented on a $\log_{10}$ scale (each dot corresponding to an individual bead, *n* = number of paired beads). **d** Proportional interaction terms of paired SC microcolonies along four categories, as schematically indicated, for the three sampling time points and for a random distribution of 1000 points in the same $\log_{10}$ area. Magenta, fraction of beads with non-growing or single cell pairs ($\log_{10}$ PBP < 3.25); red, fraction of beads with one SC partner not growing; blue, more or less equal productivity of either pair (PBP ratio between 0.125 and 8); grey, remaining pairs (moderate growth influence). Numbers within colored squares on each scatter diagram correspond to the calculated percentages of inferred interaction types for the examples in (C), and *p* indicating the probability that the observed percentages of 'no growth of one partner' is equal to a random distribution ($h_1$, alternative hypothesis; *n* = 10,000 repeats, two-sided *t*-test with unequal variance).

predictions of relative abundances than did the random distribution of growth rates (57.8% vs. 47.8% correct to within four-fold difference, Fig. 5a). This suggests that an OTU-abundance proportional $\mu_{max}$-distribution better represents SC growth, and we thus used this distribution for all further simulations. In addition, these simulations suggested that inherent growth rate differences are a predominant factor to explain the relative abundances of the SC OTUs at stationary phase in high connectivity environments, confirming our assumption of the subdued role of interspecific interactions.

Next, we simulated SC growth in a low connectivity environment, with an assumed 75% of cases of single occupancy and 25% of paired cells (similar to the experimental observations in beads). In silico cells were again randomly picked from the mean observed SC $t = 0$ OTU distribution, and we simulated growth across $2 \times 10^5$ individual in silico beads that were grouped at the system level to calculate growth per time step as a function of substrate utilization (Supplementary Methods, Section 3). We first tested the effect of having a large proportion of founder cells incapable of dividing, as was suggested by propidium iodide staining, and further assumed that such cell death may either randomly occur across all taxa or have a higher probability to affect fast-growing taxa (Supplementary Methods, Section 2.3). Secondly, we simulated whether starting single occupancy in

beads disfavors growth for many taxa, which was suggested by the experimental observations (Fig. 4a, b). We included for this a single occupancy *penalty* on the assigned $\mu_{max,sp}$, by imposing the inherent $\mu_{max,sp}$ per OTU to be inversely proportional to its initial abundance (Supplementary Methods, Section 3.1). The resulting simulated microcolony size distributions at stationary phase across all beads with single occupancy were skewed to smaller sizes, when the growth penalty and a proportionally higher death at the start was applied to fast-growers. In contrast, distributions were shifter to larger sizes in absence of growth penalty or proportional cell death (Fig. 5b), or with randomly attributed cell death at start (Supplementary Methods, Section 2.3). Among these, the distributions resulting from simulations taking proportional death and single growth penalty into account were the most similar to experimental observations of microcolony PBPs (Fig. 5c). This supported the assumption that single occupancy growth in low connectivity environments (i.e., single founder SC cell per bead) is indeed penalized, and suggested further that a significant fraction of SC cells from particularly the fast-growing taxa do not divide after their removal from the soil.

**Growth of random paired soil bacteria is best explained by a mix of positive and negative interspecific interactions.** We

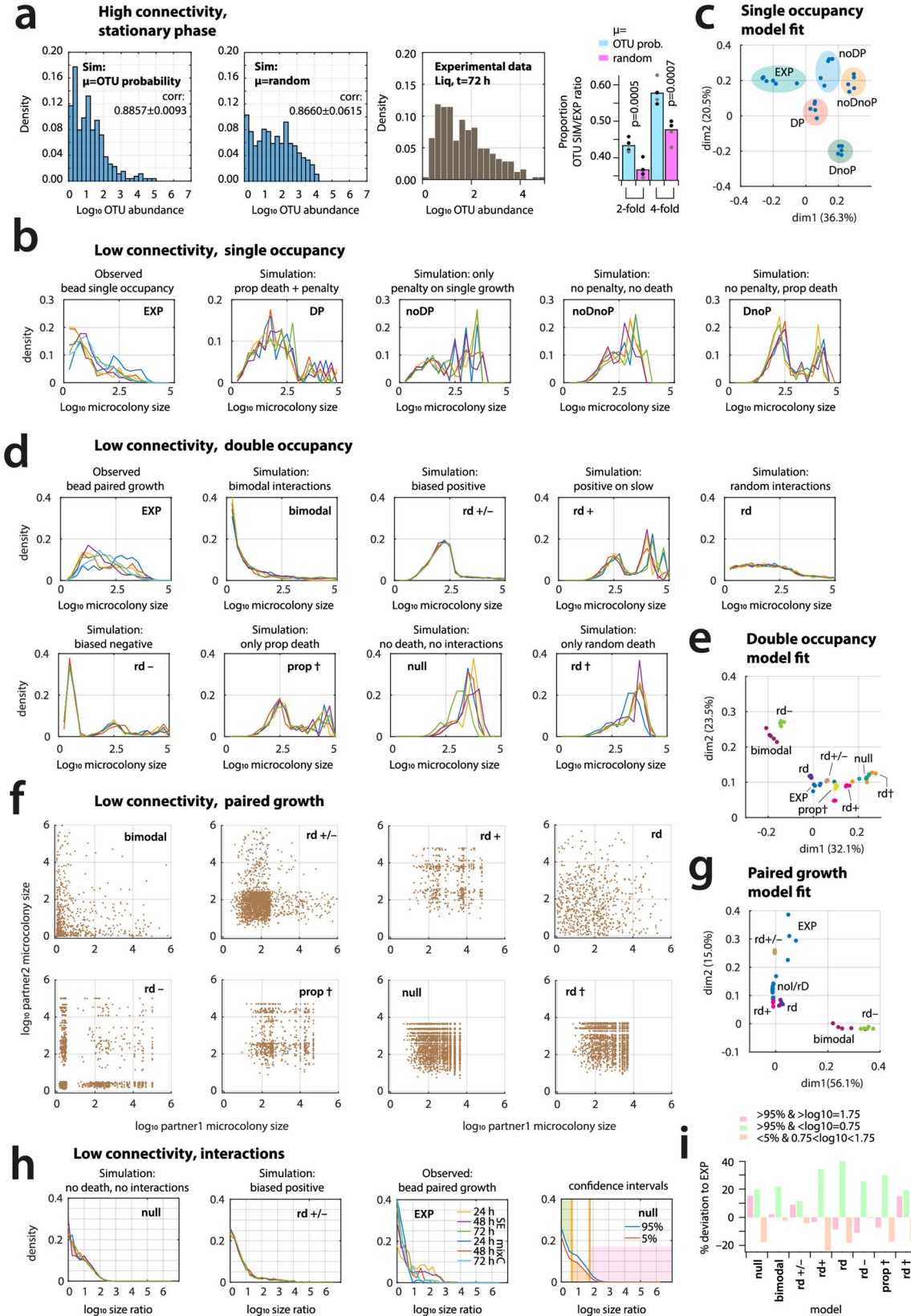

proceeded to test the hypothesis that paired growth in low connectivity environments must lead to increased importance of interspecific interactions. To this end, we simulated several interaction scenarios for the growth of OTU pairs in a low connectivity environment in comparison to a *null* model without any assumed interspecific interactions, while maintaining the same

growth rate penalty on single occupancy and the proportionally higher cell death at the start for fast-growing taxa (Supplementary Methods, sections 3.1–3.3). In the first interaction simulation scenario (Supplementary Methods, section 3.4.1, *bimodal*), we assumed that a group of taxa exists with inherently strong competitive and opportunistic character; which is suggested by

**Fig. 5 Simulations of community growth support mixed interactions prevailing low connectivity environments. a** Two scenarios for growth rate distribution among SC species, of which the one that follows the probability of OTU abundance predicts the better similarity of steady state OTU abundance distribution to observed growth in high connectivity conditions (i.e., liquid suspended growth) than random growth rate attribution. corr., mean correlation coefficient ± one SD ($n = 5$ simulations). Proportion of correctly predicted OTU abundances to within a two-fold or four-fold range of observed values (p-values from two-sided t-test, $n = 5$ simulations). **b** Observed (EXP, $t = 72$ h, $n = 6$) versus simulated steady state microcolony size abundances for low connectivity with single occupancy (i.e., one founder cell per bead, $n = 5$ simulations, each with $2 \times 10^5$ cells, subsampled to 1000 beads). DP, single occupancy growth rate penalty (proportional to the initial SC OTU abundance) plus 85% probability for fast-growing OTU to be dead at start; noDP, growth rate penalty but no dead cells at start; noDnoP, no dead, no growth penalty; DnoP, dead cells but no growth penalty. **c** Principal component analysis (PCA) of single occupancy observations and simulations (from probability normalized binned histograms), percentages showing explained variation. **d** Observed (EXP, $t = 24$, 48 and 72 h, $n = 6$) versus simulated steady state per bead normalized microcolony size abundances for low connectivity with starting OTU pairs (each with 50,000 cells, $n = 5$ simulations, in all cases in presence of 75% single occupancy beads; subsampled to 5000 beads), for seven different imposed global interaction types in comparison to null model (i.e., no death and no interactions among paired cells; Supplementary text). **e** PCA of observed and simulated data sets of paired bead growth (as for **c**). **f** Simulated paired-growth at steady state as microcolony size differences (subsampled to 5000 beads, pairs with non-growing cells removed), for the global interaction types of **d**. **g** PCA of observed and simulated paired growth (normalized from extracted events in a 12 × 12 bin grid covering $\log_{10} = 0$–6). **h** Interaction profiles (as normalized $\log_{10}$ size ratio of paired growths) for steady state low connectivity paired growth in case of the null and biased positive simulations ($n = 5$, subsampled from 5000 beads) compared to observed paired growth (EXP). **i** Percent deviation of observed paired growth ($n = 6$) to model simulations (deficit or excess compared to 5th and 95th simulated confidence intervals). Model abbreviation as in **d**.

bead productivity measurements of paired founder SC cells (e.g., Fig. 4a, b). We then compared this scenario to one of completely random interaction effects between OTU pairs (Supplementary Methods, section 3.4.4, random). Finally, we simulated three further global interspecific interaction scenarios; one that exerts generally positive effects on slow-growing OTUs in pairs (Supplementary Methods, section 3.4.3, positive on slow), one that gives a 60% chance on each partnered OTU to be positively, and 40% to be negatively influenced (Supplementary Methods, section 3.4.2 biased positive), and finally, a scenario that imposes a smaller chance on fast than on slow growers to be negatively influenced by a partner (20% and 40%, respectively; Supplementary Methods, section 3.4.5 biased negative). We let the modeled interactions directly influence the attributed OTU growth rates, thereby potentially improving the growth of a species with an initially low $\mu_{\text{max,sp}}$ or reducing that of an initially high attributed $\mu_{\text{max,sp}}$ (Supplementary Methods, sections 3.1–3.3). All models, except the bimodal, predicted that paired growth was more than two-fold more abundant than growth of single founder cells (Supplementary Fig. 4).

Simulations without any included interspecific interactions resulted in steady state microcolony size distributions skewed to the high end, which were dissimilar to experimental observations (Fig. 5d, prop†, null and rd† vs. EXP). In contrast, models assuming bimodal or biased negative interactions produced microcolony size distributions skewed too strongly to the low end (Fig. 5d). Overall, the outcomes of random and biased positive scenario simulations were the most similar to experimental observations (Fig. 5d), further attested by principal component analysis (Fig. 5e). In terms of the pattern of paired growth (Fig. 5f and Supplementary Methods, Section 3.3), the biased positive model also best described the measured combined experimental data of paired growth in beads (Fig. 5g, rd+/− vs. EXP). However, the simulations also indicated that simple growth rate differences without any interspecific interactions can result in 100–500-fold microcolony size differences among paired partners in beads (Fig. 5g, null, prop† and rd† simulations). Therefore, to estimate the potential contribution and sign (e.g., positive and negative) of interspecific interactions from observed paired growth differences (Fig. 4c, d) beyond growth rate differences themselves, we calculated and compared the distributions of paired microcolony size ratios in simulations and in experimental data (Fig. 5h and Supplementary Methods, section 3.3.5). Compared to the 95% confidence interval from five simulations, the models deviated from observations in three ranges:

underestimation of the proportion of the smallest and largest size ratios and overestimation of the mid-size ratios (Fig. 5h). The best performing simulation was taken as the one with minimal deviations in these three ranges (Fig. 5i). For example, in comparison to the null model (i.e., no interactions), the bimodal model improved prediction of low-size and mid-size, but not the large size ratios (Fig. 5i). In contrast, the biased positive model (rd +/−) deviated the least in all three ranges (Fig. 5i). This suggested that growth of random SC OTU pairs cannot be solely explained by inherent growth rate differences, but is further subject to interspecific interactions that are a non-random mixture of positive ("beneficial") and negative ("competitive") interactions, as in the biased positive model.

**Low environmental connectivity reduces community diversity almost irrespective of assumed interspecific interactions.** Finally, in order to determine the potential effects of simulated interspecific interaction regimes on community diversity, we compared simulated with observed diversity measures (e.g., Shannon index) at steady states. Measured diversities from 16S OTU assignments showed significant decreases for low and high connectivity environments compared to the beginning, and also significant differences between low and high connectivities at stationary phase (Fig. 6a). All models predicted a loss of diversity in low and high connectivity environments compared to the SC at $t = 0$ h (Fig. 6b). Importantly, the models that best explained paired growth (biased positive and bimodal) also predicted a loss of diversity in the low connectivity environment (i.e., beads) compared to the high connectivity environment (i.e., liquid growth with prop †, Fig. 6b), effectively leading to reduced taxa richness (Fig. 6c). The most realistic models predicted no increased diversity for paired growth in the low connectivity environment when compared to that across all the single occupancies (Fig. 6d), but we cannot verify this by experimental observations. Simulations explained on average 38.0% and 51.3% of individual OTU abundances within a range of two-fold and four-fold, respectively, of their observed relative abundances (SC on SE at $t = 48$ h, Fig. 6e), with Spearman rank coefficients higher for low than for high connectivity predictions, but in all cases higher than random associations (i.e., Spearman rank coefficient = 0.2385). This showed that, although we cannot predict the individual taxa behaviour very precisely, broad-scale modeling of growth rate distributions and interspecies interactions captured relevant trends of community behaviour.

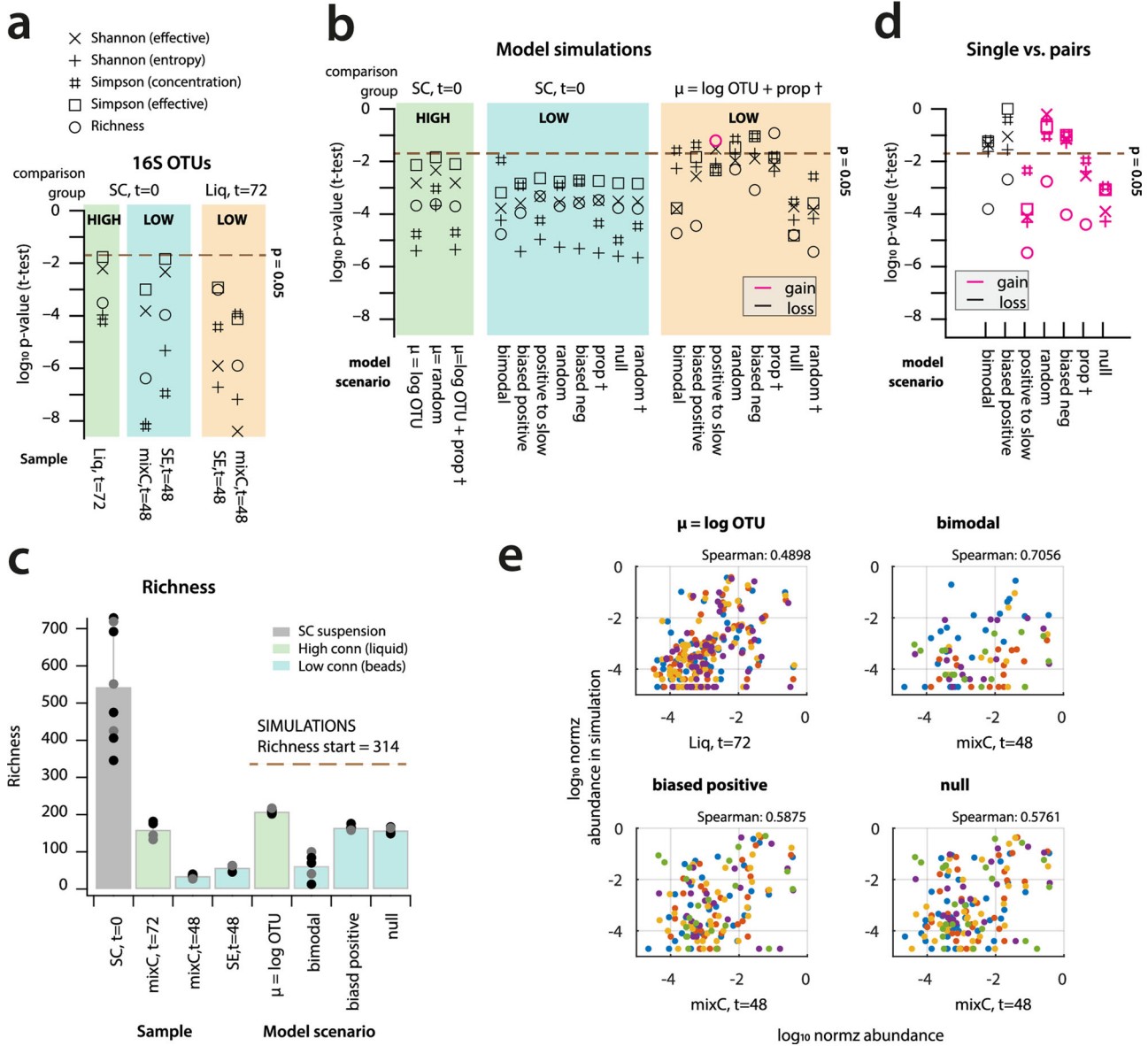

**Fig. 6 Disfavored diversity in simulated low connectivity environments. a** Log$_{10}$ probabilities of five alpha diversity measures from 16S OTU attributions being the same between comparison group and sample (bottom), expressed as *p*-value from two-sample *t*-test with unequal variance (*n* = 3–5 replicates). High, low: connectivity condition. **b** as **a**, but for simulated OTU distributions (*n* = 5 simulations, subsampled to 50,000 cells). Black symbols, loss of diversity; magenta, gain of diversity. Note how most models predict diversity loss for low compared to high connectivity environments and to starting SC diversity. **c** Observed and simulated richness decrease in high and low connectivity. **d** as **b** But for simulated diversity in single versus paired founder cells in low connectivity (five models predicting diversity gain for pairs). **e** Simulated vs. observed OTU distributions (as log$_{10}$ normalized relative abundance, *n* = 5 sets). Spearman, mean ranked coefficient of five simulations. (Spearman = 0.2385 for random normalized numbers to mixC, *t* = 48 h dataset).

## Discussion

One of the major controversies in contemporary microbial ecology is the nature of the major driving forces that determine community structure and functioning[6]. Several studies have argued that interspecific and metabolic interactions prevail[17,22], whereas others have emphasized the roles of nutrients and nutrient complexity[10,23,24], the potential to form foodwebs[25], the importance of microenvironmental changes induced by microbial activity[26], initial abundance differences[27], or the role of host factors[28]. What has received little attention is the role of connectivity of the living environment on the development of community-driving forces. Environmental connectivity should intuitively play an important role and many natural microbial communities live in environments that are characterized by

microscale patchiness, which should lead to some form of temporary disconnection from other local communities. For example, communities growing on disconnected particles (like in marine snow[29], wastewater granules[30], or as food particles in the intestinal tract[3]), or in a physically isolated environment such as a soil pore[31] or distinct plant leaf micro-habitats[32], are expected to experience degrees of disconnection. Consequently, these microbial communities will live for some time in separation, and this physical separation logically sets the boundary conditions for "other" community-driving factors (e.g., interspecific interactions) to develop.

Our results indicate that spatial separation of cells (but not of diffusible molecules) in low volume, low connectivity growth environments (agarose beads) led to a drastic decrease of

microbial diversity compared to an "all-mixed" scenario (i.e., high connectivity) of the same SC species. This is true even though community yields and aggregate growth rates were comparable for the available substrates in the system. Low environmental connectivity thus seems unfavorable for the maintenance of genotypic diversity under conditions of growth, a conclusion rooted in both experimental as well as modeling results. In contrast to many other studies, we did not bias the microbial "tester" community by using an artifical selection of laboratory strains, but recovered the starter community directly from its natural environment. This, therefore, included potentially non-viable or compromised cells or those from rare species that we could not recover to grow. Irrespective of this, however, our results clearly demonstrated that entrapment of random single or founder cell pairs in a low connectivity environment of agarose beads resulted in decreasing diversity compared to incubating the same starter community in mixed suspended cultures. We tested two different substrate regimes that both included a variety of different carbon substrates, such that we avoided any obvious nutrient selection on growth of recovered microbes. We further showed that the microbial diversity decrease is not an effect of the agarose encapsulation process itself, because samples taken after the encapsulation procedure still contained a higher diversity that gradually decreased over time.

We hypothesize that low connectivity leads to diversity loss because low connectivity environments with few founder cells amplify the role of potentially dominating interspecific interactions at this distance range[16], in comparison to high connectivity and higher starting species numbers. We could not specifically test the boundary or threshold of founding species numbers at which this effect dissipates, but the difference between 1–2 (low connectivity of agarose beads) and >300 microbial taxa (high connectivity) was clear and statistically significant. Why would interspecific interactions decrease in magnitude at higher species starting numbers? One explanation is non-transitive dynamics: the paper-rock-scissor concept[33], which stipulates that the probability that one species dominates another decreases when there is a third species that influences the first. Also, physically speaking, the chance for two negatively-influencing cells to find themselves close together for prolonged periods of time is smaller in high than in low connectivity environments. On the other hand, our results showed that microbial diversity reduced even in the high connectivity environment compared to that at start, which might be due to a low level of remaining interspecific interactions. Model simulations indeed suggested that some 60% of observed individual OTU abundances can be explained from differences in growth rates, whereas the remainder variation might be the result of interspecific interactions developing even under high connectivity by largely abundant community members.

Intriguingly, both experimental and modeling results further indicated that—at least for the natural soil community we studied here, cells on average still proliferated better being in pairs in a low connectivity environment than being alone. This suggests that many species in the starting SC community may have been auxotrophic and unable to grow alone, which is supported by other studies and hypotheses[27,34]. Their survival and growth is then on average positively influenced by the presence of other random species at short distance[16], which may produce different necessary growth factors[35–38]. Interalia we observed growth of soil-abundant but notoriously difficult to isolate species from the Acidobacteria groups[39] under high connectivity conditions. The Great Plate Count Anomaly of non-culturability[40,41] of many species detected in communities by sequencing may thus be in part due to their dependencies on one or more other species from the same community. Communities such as those in soil have

been selected for living in highly fractured environments[13,18] and their high degree of species interdependencies may thus be a consequence of co-evolution. This, again, may seem likely because negatively dominating interactions would be disfavored in small poorly connected environments. An interesting question remains, however, why microbial communities in the soil are among the most highly (species) diverse if low connectivity would tend to reduce diversity. One explanation for this conundrum might be that confined states in the soil are only temporary. Soil pores are likely to become connected after rainfall events, leading to a migration of microbial cells between previously disconnected space[42]. Mixing by earthworms and other soil invertebrates[43] and transport by fungal hyphae[44] may also help to temporarily connect small scale niches and mix new microbial constellations. Alternatively, the physical and chemical heterogeneity of low connectivity environments in soil[19,45,46] may promote more diversity, which we kept homogenous in our experiments.

What can we conclude about the potential nature of the prevailing interspecific interactions in the soil community? It has been theorized that social interactions in communities would be prevailingly negative except among closely related genotypes[47,48]. Although our experimental results indicated highly non-random imbalanced growth in partnerships, this may still be the effect of inherent growth rate differences among genotypes, and extracting the magnitude and types of interspecific interactions from complex community data is not straightforward. With an estimated ~300 different starting OTUs in the communities, there is obviously a myriad of potential interactions that can be formed. However, simulations of community growth under the assumption of different regimes of global interactions in comparison to *null* models clearly showed that growth rates alone are insufficient to explain experimental observations. Of the assumed global interspecific interactions, the *biased positive* model best explained the low connectivity experimental observations in three different test parameters (i.e., microcolony size distributions, paired growth, and OTU diversity). This suggests that interspecific interactions in the SC community are on average slightly biased to being positive, and aids the otherwise (on average) penalized single growth in isolation. Simulations further suggest that fast-growing and abundant genotypes have a higher probability to be dead in natural (freshly extracted) sand communities than slow-growing less abundant ones. What will be interesting, though, is to understand whether the globally biased interspecific interactions and penalised single growth in isolation would be something specific for a soil community or different for communities living in less fragmented environments[49]. This understanding would aid future efforts to intervene in or restore community composition in beneficial ways[50].

## Methods

**Soil resident microbes**. We chose sand as a realistic source of a mixed microbial community (which is referred to as the *sand community* or SC). Because the sand community cannot be preserved as a whole by freezing, we collected fresh material for better consistency for each experiment from the same spot in St. Sulpice near Lake Geneva (GPS coordinates: 46.508032N, 6.544050 E) as described in Moreno et al.[51]. Sampled sand at different seasons thus likely carried slightly different starting communities and cell densities. The sand was sieved through 2 mm² pores to remove large particles. The sieved sand was stored at room temperature and used within 7 days for extraction of resident microbial cells.

Microbial cells were extracted from four aliquots of 200 g of sand. Each 200 g aliquot was transferred into a 1-l conical flask and submerged in 400 ml of 21 C minimal media salts (MMS) (containing, per litre: 1 g $NH_4Cl$, 3.49 g $Na_2HPO_4 \cdot 2H_2O$, 2.77 g $KH_2PO_4$, pH 6.8)[21]. Flasks were incubated at 25 °C under rotary shaking at 120 rpm for 1 h. The sand was allowed to settle and the supernatant was decanted into a set of 50 ml Falcon tubes, which were centrifuged at 800 rpm with an A-4-81 rotor and a 5810R centrifuge (Eppendorf AG.) for 10 min to precipitate heavy soil particles. Supernatants were decanted into clean 50 ml Falcon tubes and centrifuged at 4000 rpm for 30 min to pellet cells. The supernatants were carefully discarded and the cell pellets were resuspended and

pooled from the four aliquots (i.e., from the initial 800 g of sand) in one tube using 5 ml of MMS. The pooled liquid suspension was further sieved through a 40 µm Falcon cell strainer (Corning Inc.) in order to remove any particles and large eukaryotic cells that may obstruct flow cytometry analysis (see below). A small proportion of the sieved liquid suspension was used to quantify the numbers of recovered cells (see below); the remainder was used within 12 h for bead encapsulation or for mixed liquid suspended growth (see below). With this gentle method, we extracted approximately $3 \times 10^5$ cells $g^{-1}$ of sand.

**Flow cytometry cell counting.** Cell numbers in extracted soil communities and in the mixed liquid suspended growth experiments were counted by flow cytometry. SC-suspensions were diluted 100 times in MMS and stained in 200 µl aliquots with 2 µl of diluted SYBR Green I solution (1:100 in DMSO; Molecular Probes) in the dark for 30 min at room temperature. In some experiments, cells were additionally stained with 2 µl propidium iodide solution (10 µg ml$^{-1}$, Molecular Probes). Aliquots of 20 µl were aspired at 14 µl min$^{-1}$ on a Novocyte flow cytometer with absolute volumetric cell counting (ACEA Biosciences, USA). Cells were thresholded above a forward scatter signal (FSC-H) of 20 and further gated for propidium iodide-staining (excited at 535 nm and its fluorescence was collected at 617 ± 30 nm) and for SYBR Green I (excitation 488 nm, 530 ± 30 nm band-pass filter; channel voltage at 441 V) above values of 1000 (Supplementary Fig. 5).

Cell samples from the mixed liquid suspension growth experiments were diluted to approximately $10^6$ ml$^{-1}$ and subsampled to aliquots of 100 µl. The subsamples were then mixed with 100 µl of 8 g l$^{-1}$ sodium azide in phosphate buffered saline and incubated for 1 h at 4 °C to arrest cell respiration and growth. Samples were then stained with SYBR Green I as above and quantified by flow cytometry using the same thresholds and gates as describe above.

**Bacterial strains and pre-culturing procedures.** *P. veronii* 1YdBTEX2 is a toluene, benzene, *m*-xylene and *p*-xylene degrading bacterium isolated from contaminated soil[20]. The strain was tagged with a single-copy chromosomally inserted mini-Tn7 transposon carrying a P$_{tac}$–*mCherry* cassette (*Pve*, strain 3433) as described in the ref. [52]. A single *P. veronii* colony from a selective plate with toluene as the sole carbon substrate after 48 h incubation at 30 °C was inoculated into 10 ml of liquid MMS containing 5 mM sodium succinate as the sole carbon source and grown for 24 h at 30 °C with rotary shaking at 180 rpm[21]. After 24 h, the cells were harvested and washed for bead encapsulation or for comparative liquid mixed suspension growth, as described below.

**Agarose bead encapsulation.** SC cell suspensions containing between $2 \times 10^7$ to $10^8$ cells ml$^{-1}$ were encapsulated in agarose using rapid mixing with pluronic acid in dimethylpolysiloxane and subsequent cooling, followed by sieving to achieve beads with a diameter range of 40–70 µm[53]. The entire procedure was carried at room temperature and near a gas flame to maintain antiseptic conditions. 1% (*w/v*) low melting agarose (GEPAGA04-62, Eurobio ingen, France) was prepared in PBS solution (PBS contains per L H$_2$O: 8 g NaCl, 0.2 g KCl, 1.44 g Na$_2$HPO$_4$, 0.24 g KH$_2$PO$_4$, pH 7.4) and dissolved by heating in a microwave. The molten agarose solution was cooled down and equilibrated in a 37 °C water bath. Separately, 15 ml of dimethylpolysiloxane (Sigma-Aldrich, DMPS5X-500G) was poured in a 30 ml glass test tube. 1 ml of the 37 °C-agarose solution was mixed with 30 µl of pluronic acid (10% Pluronic® F-68, Gibco, Life Technologies) by vortexing at the highest speed (Vortex-Genie 2, Scientific Industries, Inc.) for a minute. Into this mixture of agarose and pluronic acid, 200 µl of prepared SC cell suspension at $0.2–1.0 \times 10^8$ cells ml$^{-1}$ was pipetted and vortexed again at the highest speed for another minute. Five hundred microliter of this mixture was added drop-wise into the glass tube with dimethylpolysiloxane that was being vortexed at maximum speed. Vortexing was continued for 2 min. The tube was then immediately plunged into crushed ice and allowed to stand for a minimum of 10 min. After this, the total content of the tube was transferred into a 50 ml Falcon tube. The tube was centrifuged for 10 min at 2000 rpm using an A-4-81 swinging-bucket rotor (Eppendorf). The oil was carefully decanted while retaining the beads pellet. Fifteen milliliter of sterile PBS was added to the pellet and the beads were resuspended by vortexing at a speed set to 5. The tubes were again centrifuged at 2000 rpm for 10 min and any visible oil phase on the top was removed using a pipette. The process was repeated once more to remove any visible oil phase. Beads of diameter between 40 and 70 µm were then recovered by passing the PBS-resuspended bead content of the tube first over a 70-µm cell strainer (Corning Inc.). A further 5 ml of PBS was added to the cell strainer to flush remaining beads (<70-µm) into the filtrate. The collected bead filtrate was subsequently passed over a 40-µm cell strainer (Corning Inc.) to remove beads smaller than 40 µm. Recovered beads on the sieve were washed with an additional 5 ml of PBS, and any smaller beads in the filtrate that stuck to the bottom side of the cell strainer were gently removed by absorption with a Whatman 3M filter paper. After this, the sieve was inverted and placed on top of a clean 50 ml Falcon tube. 1.5 ml of incubation medium (MMS with the respective carbon substrates, see below) was used to collect the beads from the sieve into the tube. Two tubes were processed in parallel, which were pooled in the same final Falcon tube to yield a total volume of 3 ml. This volume was then split in three aliquots of 1 ml to make triplicate incubations. The encapsulation procedure produced ~$1.2 \times 10^6$ beads per ml, with an effective volume of 10% of the total volume of the liquid

phase in the incubations. For comparative growth experiments, the procedure was repeated with the *P. veronii* pure culture. A visual guide to the procedure is presented in Supplementary Methods, Section 4.

**Cell-in-bead growth incubations.** Each of the incubation vials (with 1 ml bead solution) was further complemented with 4 ml MMS containing either mixed carbon substrates ("Mixed-C", 0.1 mM C) or sand extract (see below) as growth substrates. Mixed-C solution was prepared by dissolving 16 individual compounds (Supplementary Table 2) in milliQ-water (Siemens Labostar) in equimolar concentration such that the total carbon concentration of the solution reached 10 mM C. These compounds are also listed in EcoPlates$^{TM}$ (Biolog Hayward, CA, USA) and have been previously used as a soil representative substrates[54].

Sand extract was prepared by extraction with pre-warmed (70 °C) sterile milliQ-water. A quantity of 100 g sand was mixed with 200 ml milliQ-water in a 250 ml Erlenmeyer flask and swirled on a rotatory platform for 15 min, after which it was subjected to 10 min sonication in an ultrasonic bath (Telesonic AG, Switzerland). Sand particles were sedimented and the supernatant was decanted, and passed through a 0.22-µm vacuum filter unit (Corning Inc.). This formed the "sand extract", of which 4 ml was added directly to the 1 ml bead suspension in the vials.

At the amended substrate concentrations, on average between 8 and 18% of SYTO-9-stained cells were detected on images outside (automatically detected) beads. As these proportions did not increase over time, we conclude that these were not cells growing outside beads but cells escaped from beads squeezed and broken between the microscope slide and coverslip during imaging. We therefore did not further take them into consideration when judging intrabead cell growth.

**Mixed liquid suspension growth.** For growth of SC-cells or of *P. veronii* in regular mixed liquid suspension, 2 ml of MMS with 0.1 mM mixed-C substrates were inoculated in quadruplicate with $5 \times 10^5$ cells ml$^{-1}$ (prepared as described above for bead encapsulation and growth). As a negative control for background growth, MMS without added carbon substrate was inoculated. Assays were incubated at 25 °C with rotary shaking at 120 rpm, sampled daily (100 µl) for cell fixation, staining and flow cytometry counting of SC-cell numbers.

**Bead sampling and microscopy.** For sampling, the vials were removed from the incubator and beads were collected at 1200 rpm for 1 min using a swinging-bucket A-4-81 rotor (Eppendorf). An aliquot of 10 µl of bead suspension was carefully sampled from the bottom of the vials, mixed with 0.6 µl of 50 µM SYTO-9 solution to stain cells, and incubated for 20 min at room temperature in the dark. Vials were vortexed and placed back into the incubator. Five microliter of sterile milli-Q water was added to the stained beads and the complete aliquot (15 µl) was spread on a regular microscope glass slide to minimise aggregation of beads. A coverslip (24 × 50 mm) was gently placed to avoid air bubbles and excessive squeezing of the beads. Ten random positions on the slide were imaged with the 20× objective (NA 0.35) using an inverted AF6000 LX epifluorescence microscope system (Leica AG, Germany) equipped with a DFC350FXR2 camera. Every position was imaged in four sequential channels (phase contrast, 25 ms; mCherry, Y3-cube, 750 ms; SYTO-9, GFP-cube, 50 and 340 ms). The 50 ms-SYTO-9 channel exposure was used for analysis while the 340-ms exposure was used for verification of weak signals, if necessary. Images were recorded as 16-bit TIF-files and further processed using a custom MATLAB routine (described below).

**Microscopy image analysis.** A custom MATLAB image processing and analysis routine was developed to segment beads and microcolonies inside beads from the image-series[55]. For each time-point and experimental replicate, the phase contrast, mCherry, and SYTO-9 images were read using the *imread* function built in MATLAB (version 2016b, MathWorks inc., USA). To identify the beads on each image, sharp changes in intensity were detected in the phase-contrast images using the *edge* function. Individual beads within a specific radius range were then identified using the *imfindcircles* function. In the next step, the microcolonies inside each bead were identified by thresholding and segmenting the mCherry and SYTO-9 images exclusively within the identified bead areas. mCherry and SYTO-9 images were further aligned to identify microcolonies in SYTO-9 having mCherry signal, which corresponds to *Pve*. Overlapping signals were considered to originate from a *Pve* colony if the area overlap between two channels was at least 30% or larger. If this were not the case, the areas were considered to consist of both *Pve* and SC microcolonies. Object areas smaller than 2 pixels were discarded. All microcolonies were thus differentiated as corresponding to *Pve* (mCherry plus SYTO-9 signal) or SC (SYTO-9 only), after which their area, fluorescence intensity and geometric distance (within the bead) were calculated.

Results were summarized for each incubation and time point to comprise the following information: (i) total number of beads containing microcolonies; (ii) the mean per bead productivity (PBP) (the product of the identified areas times their mean SYTO-9 fluorescence intensity, normalized over all analyzed beads); (iii) the normalized PBP distribution and its bin sum; and (iv) the PBPs for beads with single SC-cell occupancy or with SC-cell pairs.

**Community diversity analysis.** The diversity of the starting and growing SC communities was analyzed by high-throughput sequencing of amplified V3–V4

regions of the 16S rRNA gene. DNA was isolated from cell pellets from 1.5 ml liquid SC suspensions or from beads (2 ml) kept at –80 °C until analysis using a FastDNA Spin kit for soil according to the manufacturer's instructions (MPBio). We targeted the V3–V4 hypervariable region of the 16S rRNA gene by amplification with the 341f/785r primer set and appropriate Illumina adapters and barcodes, following recommendations of the reagent supplier (https://support. illumina.com/documents/documentation/chemistry_documentation/16s/16s-metagenomic-library-prep-guide-15044223-b.pdf). The 16S Amplicon PCR Forward Primer = 5′

TCGTCGGCAGCGTCAGATGTGTATAAGAGACAGCCTACGGGNGG CWGCAG. The 16S Amplicon PCR Reverse Primer = 5′

GTCTCGTGGGCTCGGAGATGTGTATAAGAGACAGGACTACHVGGGT ATCTAATCC. Equal amounts of amplified DNA from each sample were pooled and sequenced bidirectionally on the Illumina MiSeq platform at the University of Lausanne Genome Technologies Facilities. Raw 16S rRNA gene amplicon sequences were quality filtered, concatenated, verified for absence of potential chimeras, dereplicated and mapped to known bacterial taxonomy (OTUs) using QIIME2 at 99% similarity to the SILVA taxonomic reference gene database on a UNIX platform[56]. OTU-assigned reads were normalized to the sum of reads per sample and plotted by multi-dimensional scaling based on calculated Bray–Curtis distances as implemented in the *R phyloseq* package (v1.16.2).

**Community growth modeling in high and low environmental connectivity**. We developed a custom model (MATLAB) to test the effects of environmental connectivity and population growth on community diversity in presence or absence of interspecific interactions. Simulations were used to derive OTU diversity at stationary phase (OTU distributions, paired-growth, and diversity measures). Detailed description of the model is provided in Supplementary Methods, Sections 1–3. Code examples are available[55]. Briefly, we modeled community growth from 200,000 in silico cells at the beginning (corresponding to the inoculated $2 \times 10^5$ cells ml$^{-1}$ in the experiments), either in a high connectivity (as in our liquid experiments) or low connectivity (as in the agarose bead-encapsulated cells) evironment. In the latter case, we simulated both single (75% of all in silico beads) or pairs of founder cells (25%). Single cells were simulated as a one-dimensional vector with length of 200,000 and pairs as two one-dimensional vectors ($2 \times 200,000$).

The initial OTU composition of the vector was derived from the experimentally determined OTU distribution from 16S rRNA amplicon sequencing in the washed soil suspensions before inoculation. The probability of occurrence of each OTU is computed as:

$$P(OTU_i) = 100 \times \left( \frac{OTU_i}{\sum_{1}^{n} OTU_i} \right) \qquad (1)$$

and was used in sampling the 200,000 initial cells at $t = 0$ h (Supplementary Methods, Section 1). To simulate the dynamics of growth, we employed Monod kinetics, described as:

$$\mu_{sp,i} = \mu_{max,sp,i} \frac{S}{K_s + S}, \qquad (2)$$

where $\mu_{sp,i}$ and $\mu_{max,sp,i}$ denote the (maximum) specific growth rate of species $i$, $S$ is the prevailing substrate concentration, and $Ks$ is the concentration of the substrate at the point, where the specific growth rate is half of the maximum growth rate. All growth kinetic parameters except the growth rate were kept the same for all species. For simplicity and in absence of empirical kinetic data on SC cells, we assumed a general $K_S = 0.3 \times 10^{-6}$ g ml$^{-1}$, constant yield = 0.3 carbon to biomass conversion in g g$^{-1}$ and constant mass = 120 fg cell$^{-1}$ for all species. The yield factor accounted for $CO_2$ losses from carbon metabolism. In contrast, we varied $\mu_{max}$ between 0.01 and 0.6 h$^{-1}$, and conditionally included *growth penalty* factors on single founder cells in the low connectivity environment (as explained below). We further examined the effect of having genotype-independent (randomly attributed) death at the beginning or death biased to fast growing genotypes ($\mu_{max} > 0.25$). Finally, we tested further *interaction terms* that influenced attributed growth rates. The initial carbon concentration was set to 50 mg ml$^{-1}$, which allowed similar community development in terms of size (i.e., cell numbers) as in the experiments.

Growth was simulated for 120 time steps, corresponding to 60 h in the experiments, at which point the substrate is depleted and cells stop dividing (stationary phase). Based on the attributed growth rates to every OTU (i.e., every cell and genotype of the vector or double vector), the model calculates per time step how much substrate is converted into biomass (we allow continuous biomass formation) and lost in form of $CO_2$, which is subtracted to calculate the remaining substrate concentration for the next time step. When the overall substrate concentration is lower than $S_{min} = 3 \times 10^{-6}$ g ml$^{-1}$, growth stops. The production of cell biomass is converted to cell numbers, which is then subsampled at the last time step per OTU (to an equivalent of 50,000 sequence reads), per single or pair (to an equivalent of 5000 beads) to calculate developed microcolony sizes, diversity measures, and interaction effects (as in Figs. 5 and 6).

The following interaction scenarios were simulated. Although we allowed growth penalties and interspecific interactions to influence attributed OTU growth

rates, a threshold of ~0.6 h$^{-1}$ was imposed as the maximum individual OTU growth rate in all simulations.

*High connectivity random vs. OTU-abundance growth rates.* OTU-specific growth rates ($\mu_{max,sp}$) were drawn randomly between 0.01 and 0.6 h$^{-1}$. Alternatively, growth rates were attributed to the vector of OTUs according to the probability distribution function reflecting their measured log$_{10}$ empiric abundance at $t = 0$ h (Supplementary methods, Section 2).

*Low connectivity single founder cell growth penalty.* We contrasted simulations with single founder cells growing according to their OTU-proportional attributed growth rate and those in which that growth rate was multiplied by a penalty, composed by a factor equal to the inverse proportion of the initially attributed $\mu_{max,sp}$ per OTU. The assumptiton is that the slower the inherent growth rate, the more likely that OTU is penalized when it is alone (Supplementary methods, Sections 2 and 3). This was combined with testing the effect of random or biased death on the starting community.

$$\mu_{max,sp} = \frac{1.2}{\log_{10}(\mu_{max,sp})} \qquad (3)$$

*Low connectivity paired interspecific interaction effects.* We further tested different assumptions on the nature of interspecific interactions and simulated how these affected community growth rates and diversity outcomes. These effects directly influenced the OTU attributed growth rates in the *doubles* (Supplementary methods, Sections 3.1–3.3). In the *bimodal* scenario (Supplementary methods, Section 3.4.1), we assumed that the community is composed of two underlying distributions: rare and abundant members (the threshold being placed at log$_{10}$ measured OTU relative abundance = 2.8), with abundant members having a higher probability to be positively influenced in pairs. The probability is drawn from a bimodal interaction curve that attributes an interaction factor (between 0.01 and 2.2), which is multiplied with the assigned OTU growth rate at start.

In the *biased positive* model (Supplementary methods, Section 3.4.2) we allowed a 40% chance for an interaction term imposed independently on each founder cell in a pair to lower the attributed OTU growth rate (factor range 0.4–0.6), and 60% chance for a factor in between 0.6 and 1.4 to modulate or increase the growth rate.

In the *positive on slow* model (Supplementary methods, Section 3.4.3), the attributed OTU growth rates on each founder cell in a pair had a chance of 40% to become improved inversely proportionally to its initial growth rate, thereby favoring slow growers

$$\mu_{max,sp} = -\ln(\mu_{max,sp}) \times \mu_{max,sp} \qquad (4)$$

In the *biased negative* model (Supplemtary methods, Section 3.4.4), we attributed OTU-abundance proportional growth rates to each partner of the founder pair, but penalized faster growers ($\mu > 0.15$) at 20% chance and the others at 40% chance that their growth rate would be multiplied by a negative interaction factor (range 0.01–0.1).

Finally, in a *random* model (Supplementary methods, Section 3.4.5), we allowed OTU-abundance proportional growth rates in pairs to be multiplied with a factor randomly drawn in the range of 0.01–1.25, independently for each partner in a pair. The models were contrasted to those without any assumed interspecific interactions, and without or with assumed random or fast-growing genotype biased cell death at start (Supplementary methods, Section 2.3.1).

All simulations were run five times from the beginning, independently producing five derived parameter values for alpha-diversity, OTU- and microcolony size distributions in stationary phase and partner interactions.

**Statistics and reproducibility**. Liquid suspension growth experiments were carried out in biological quadruplates and all bead experiments were carried out in biological triplicates. Total numbers of analyzed bead and those of beads with single or double occupancy are reported. Derived community growth rates and *P. veronii*-normalized yields were compared using *t*-tests (*n* as reported, two-sided test, unequal variance). Normalized PBP bin-size distributions were globally compared using Fisher's exact test implemented in R (2000 replicates). Median and 75th percentile aggregate PBPs across different experiments were compared using the non-parametric Wilcoxon signed-rank test. Correlations between simulated species abundance distributions and empirical OTU relative abundances were calculated by bootstrapping (*n* = 1000) in MATLAB. Correlation coefficients from five independent simulations were compared using *t*-tests. The proportion correctly predicted OTU abundances by simulation was calculated as the ratio to observed values within a two-fold or four-fold range, and compared by two-sided *t*-tests on five independent simulations. Simulated and observed microcolony size distributions for single or paired founder cells among different models were compared by principal component analysis in MATLAB (*pca*), and by Spearman rank correlation (*spear*) from five independent simulations. Single and paired productivity was compared between each other using two-sided *t*-tests of the 75th percentiles of microcolony size distribution (*n* = 5). Simulated and observed paired growth (excluding pairs with dead cells) was categorized and counted in a grid of 12 × 12 (each bin covering 0.5 log$_{10}$-distance) using MATLAB's *hist3d* function, and then compared by *pca* from five independent simulations. Confidence intervals on ratios of paired simulated microcolony sizes (excluding those pairs with a non-growing or dead partner) were determined by subsampling

($n = 1000$) from mean ratio distributions, which were then used to calculate the fractions deviating from experimentally observed paired size ratios.

**Reporting summary**. Further information on research design is available in the Nature Research Reporting Summary linked to this article.

## Data availability
The raw data for the 16S rRNA amplicon sequencing data can be accessed from the Short Read Archives under BioProject ID PRJNA661487. All source data are available as Supplementary Data in Excel format. Please see Description of Additional Supplementary Files for more information.

## Code availability
MATLAB codes for modeling and image analysis (version 2016a) are available for download without any restriction from Zenodo[55].

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

## Acknowledgements

This work was supported by SystemsX.ch, and evaluated by the Swiss National Science Foundation, within grant 2013/158 (Design and Systems Biology of Functional Microbial Landscapes "MicroScapesX"), and by the National Centre in Competence Research in *Microbiomes*. We thank Robin Tecon and Ljubisa Miskovic for critical reading and feedback.

## Author contributions

M.D. and N.C. provided experimental data. S.P. and N.H. developed the initial scripting. M.D., N.H., and J.R.M. revised and improved community models, and validated data comparisons. M.D., N.C., N.H., and J.R.M. analyzed data. D.J. advised statistical testing. M.D., N.H., D.J., and J.R.M. wrote the main text. All authors corrected and approved the final text.

## Competing interests

The authors declare no competing interests.
