## [Peer Review File · Communications Biology]

Reviewers' comments:

Reviewer #1 (Remarks to the Author):

The authors investigate the effects of environmental connectivity on community growth and microbial diversity. They contrasted growth of a sand microbial community with high species diversity in two environments: growth in a well-mix liquid suspension (high connectivity environment) and growth in separated agar beads (low connectivity environment). The miniature agarose beads containing random single or paired founder cells at the start of the experiments, which then grow and are analysed at different times, to assess biomass production and system-level species diversity. The authors found that growth was the same in the two growth condition (i.e. low and high connectivity environments), but that low connectivity environment reduced community diversity. The authors explain their experimental observations by modeling community growth and interspecies interactions. They suggest that microbial diversity loss is the result of negative interspecific interactions becoming more dominant at small founder species numbers in low connectivity conditions.

The work is conceptually interesting and the experimental results are sound. The experiments are described carefully, in a way that allows others to reproduce them. My main concern is about the mechanisms the suggest for explaining their experimental observations. I find that the role of non-growing cells should be analysed better. It is possible that non-growing cells could explain the experimental observation as well as interspecies interactions, which is the mechanisms suggested in the manuscript. I describe this point in the "Major comments" section. Finally, the manuscript could benefit from more precision in the terminology, as indicated by my minor comments.

*****Major comments*****

I am concerned about their conclusion that growth is disfavored when being alone. The author summarizes this conclusion at line 241: "Collectively, these observations showed that cells from a highly taxa-rich microbial community from soil, randomly placed in a multitude of parallel low connectivity environments are on average growth disfavored when being alone and favored when being partnered, even though the majority of developed interactions are negative on one of the partners."

It seems to me that the experimental observations are consistent both with the above interpretation and with the eventuality that only some cells (randomly) start growing, while other not. In this second eventuality, when cells are encapsulated in pairs, there is double chance of having in the bead one of the cells that start growing. My concern is supported by the widespread observations that, within the same species, cells display heterogeneity in their ability to respond to environmental challenges (as it is the one posed by encapsulation in agar beads). See for example: [Dal Co et al. J.Royal Soc Inteface 2019, <https://doi.org/10.1098/rsif.2019.0182>], and [Schreiber et al. Nat. Microbiol. 2016, doi:10.1038/nmicrobiol.2016.55]). Can the author exclude that their results could be explained by heterogeneity in the chance to grow? Could they perhaps use their model to explore this eventuality? E.g. the model could have a random probability of single cells to restart growing.

My concern also applies to the statement at line 202: "In addition, only 19.7% (95% confidence interval 14.1–25.3) of beads with single occupancy surpassed a 10-fold productivity increase compared to the start, versus 47.4% (37.5–57.3, n = 12) for beads with two or more microcolonies. This indicated that being randomly partnered was on average beneficial for productivity.."

Is the result described here also consistent with the scenario where only one of the two randomly paired cells can grow? If only ~20% of species can grow in isolation up to 10 folds, then pairing them could lead to ~40% pairs growing to 10 folds, but where once of the two pairs doesn't grow at all.

Finally, the author could check if the fact that cells could randomly not grow would able to explain the loss of diversity. In my belief, the authors discuss too quickly the role of non-growing cells; see line 383 : "Alternatively, since we worked with cells from natural communities, the estimated

proportion of 20–30% non-dividing SC founder cells may have skewed community diversity upon regrowth.”

Overall my concern is that their experimental results can arise from alternative mechanisms than what the authors suggest and explore with their model. I think these alternative mechanisms w=should be dismissed.

****Minor comments****

75 “the less likely that they may dominate the forming microcommunity of cells” unclear if “they” refers to species or to interactions.

82 “a sort of egalitarianism at high species numbers” what do the authors mean with this? They first mention challenges, limitations and selection. What does it mean that these challenges, limitation and selections become egalitarian when there are multiple species? Can the author rephrase more specifically what they mean here?

94 I believe that the authors should comment here about the diffusibility across agar beads. The introduction suggests that “low connectivity” refers not just to no cell movement but also to no diffusion between microenvironments.

100 what do they mean with “deduce the scale and global nature of interspecific interactions”? I think here the formulation “scale and global nature” is not clear.

106 Can the author elaborate or rephrase more directly the sentence “Despite expected increasing dominance of interspecific interactions, randomized partnerships were beneficial for community growth as opposed to growth in complete isolation.” Are the authors saying that, despite competitive interactions, randomize partnerships were beneficial compared to isolation?

Do soil bacteria degrade agarose? I thought many did (e.g. see: Chi, Applied Microb. And Biotech, 2012, <https://link.springer.com/article/10.1007/s00253-012-4023-2>). Would consumption of agarose in one of the two environmental conditions (the low connectivity condition) affect the authors’ main conclusions?

232 it’s not clear to me why higher differences would reflect faster and more abundant growth. The text says: “higher magnitude of microcolony PBP differences than on mixed C substrates, reflecting the faster and more abundant growth on that substrate”

352 the authors should talk about diffusible molecules also earlier, I believe.

587 “Overlapping signals were considered to originate from a Pve colony if the area overlap between two channels was greater than 30%.” Why 30%? That sounds like a low threshold to me.

588 for the first time here “particle” is used to refer to a group of cells inside the bead. Before “particle” is present for various use. I’d suggest using another word other than particle. This was unclear to me.

897 “the mean per bead productivity (PBP), or the product of the particle pixel area times its SYTO-9 fluorescence intensity normalized over all analyzed beads;” Two points: 1) I don’t understand why the productivity is calculated as an area x bead intensity. I assume it should be area x mean intensity. 2) how does the area increase related to the volume increase, which is the actual real measure of biomass increase? How do they go exactly from their area and intensity measure to a volumetric biomass estimation? Can they explain more clearly this calculation also in the supplementary text?

665 typo “slowed” would be “slower”. English should be revised here.

680 “and 60% chance for a factor in between 0.8–1.4 that could increase the growth rate.” How can the factor 0.8 increase the growth rate? Did I miss something?

940 and other occurrence in main text and SI: can the authors write \log_{10} PBP more precisely? Where 10 is clearly the base of the logarithm?

Fig 5b μ seems to be not defined

Fig 5d there is no reference in text to this panel.

Fig 5e there is a typo on the numbers on x axis

Fig 5f I find the heat map (right) hard to read. For example, what should the reader see about the correlations between models? I'd consider making a simpler figure, that conveys the message the authors wanted to convey with this more complicated figure. Note also that Fig 5f (left plot) has axis with no units.

Reviewer #2 (Remarks to the Author):

In this manuscript, the authors explore and investigate the role of environmental connectivity in shaping community growth and species diversity and underlying interspecific soil microbial interactions. In particular, through laboratory experiments, they studied the growth of naturally-derived soil microbial communities and demonstrate that low environmental connectivity appears to reduce global community diversity as compared to high connectivity conditions. In addition, via computational community growth modeling, they further demonstrate that this microbial diversity loss seems to result from biased negative microbial interactions. Their results lend support to the importance of interspecific interactions in shaping natural microbial communities, which need to be considered and fundamentally understood to move towards their control in medicine and biotechnologies.

This work combining both experiments and modeling to investigate the influence of environmental connectivity on microbial community growth and diversity is relevant, well designed, and novel. Overall, their analyses are pertinent and sound, the results are well presented, and the manuscript is well written. Overall, I have only minor comments, which are detailed below. Thus, I think this manuscript should be considered for publication in *Communications Biology*, once the authors have successfully addressed the recommended revisions.

Minor comments:

- The authors studied the effect of environmental connectivity on the growth and diversity of soil microbial communities. Although their results may potentially apply to microbial communities from other biomes (e.g. host-associated, marine), it is not demonstrated in the present study. Thus, I would recommend, where appropriated (for example in the title), to specify that the results and conclusions presented in the manuscript apply only to soil microbial communities.
- Different methodologies were used to quantify biomass in low versus high connectivity conditions. Although comparing SC yields as the ratio to *P. veronii* is appropriate, it is biased by the actual growth ability of this specific species in both conditions. Thus, a direct comparison of relative biomasses through time in both conditions may be complementary to their approach.
- Results presented in Fig. 3a may benefit from a more comprehensive representation of all experimental results (through time) by comparing beta-diversity across all samples, for example via a Principal Coordinates Analysis. This would allow visualizing the evolution of the global community compositions, and how they relate to each other along the course of the experiments.
- In the discussion, the authors state the hypothesis "that low connectivity leads to diversity loss because low connectivity environments can contain fewer starting cells, ..., in comparison to high connectivity and higher starting species numbers". Here, the authors assume that low connected environments may contain fewer cells, but I must say I am not sure to understand the rationale behind this assumption. Since (soil) microbial communities are highly dynamic, a low connected environment may also contain a high diversity. Thus, I suggest the authors clarify and elaborate on this hypothesis.

Rebuttal letter

Please note that line numbers here refer to the PDF version of the manuscript with all changes highlighted (to avoid confusion between old and revised text).

Referee expertise:

Referee #1: systems biology, microbial ecology

Referee #2: computational biology, microbial ecology and evolution, systems ecology

Reviewers' comments:

Reviewer #1 (Remarks to the Author):

The authors investigate the effects of environmental connectivity on community growth and microbial diversity. They contrasted growth of a sand microbial community with high species diversity in two environments: growth in a well-mix liquid suspension (high connectivity environment) and growth in separated agar beads (low connectivity environment). The miniature agarose beads containing random single or paired founder cells at the start of the experiments, which then grow and are analysed at different times, to assess biomass production and system-level species diversity. The authors found that growth was the same in the two growth condition (i.e. low and high connectivity environments), but that low connectivity environment reduced community diversity. The authors explain their experimental observations by modeling community growth and interspecies interactions. They suggest that microbial diversity loss is the result of negative interspecific interactions becoming more dominant at small founder species numbers in low connectivity conditions.

The work is conceptually interesting and the experimental results are sound. The experiments are described carefully, in a way that allows others to reproduce them. My main concern is about the mechanisms the suggest for explaining their experimental observations. I find that the role of non-growing cells should be analysed better. It is possible that non-growing cells could explain the experimental observation as well as interspecies interactions, which is the mechanisms suggested in the manuscript. I describe this point in the "Major comments" section. Finally, the manuscript could benefit from more precision in the terminology, as indicated by my minor comments.

1. Reply: *We thank the reviewer for the general appreciation and the positive constructive suggestions for improvement.*

We agree that we had not specifically accounted for cell death in the modeling steps, although we did measure the proportion of potential dead cells in the starting cell community suspensions by propidium iodide staining (l. 170). The reviewer was thus right that this could potentially explain part of our observations, particularly of figure 4c and d.

Action: *We completely revised the community growth model and tested specifically for the inclusion of cell death, in two ways: (i) a random equal chance for all species to be dead at the time of start of the experiment, and (ii) a death rate overproportionally high for fast-growing*

species. This second assumption explained better the observed experimental behaviour and was then included in all subsequent interaction models. We also included subsampling from the modeled species growth distributions to better account for the limited experimental analysis (i.e., sequencing down to a depth of 50,000 reads per sample replicate, and 1000 beads per replicate). We clarified this in a complete overhaul of Figure 5, a new Figure 6 and the revised supplementary method sections.

*****Major comments*****

I am concerned about their conclusion that growth is disfavored when being alone. The author summarizes this conclusion at line 241: "Collectively, these observations showed that cells from a highly taxa-rich microbial community from soil, randomly placed in a multitude of parallel low connectivity environments are on average growth disfavored when being alone and favored when being partnered, even though the majority of developed interactions are negative on one of the partners."

It seems to me that the experimental observations are consistent both with the above interpretation and with the eventuality that only some cells (randomly) start growing, while other not. In this second eventuality, when cells are encapsulated in pairs, there is double chance of having in the bead one of the cells that start growing. My concern is supported by the widespread observations that, within the same species, cells display heterogeneity in their ability to respond to environmental challenges (as it is the one posed by encapsulation in agar beads). See for example: [Dal Co et al. J.Royal Soc Interface 2019, <https://doi.org/10.1098/rsif.2019.0182>], and [Schreiber et al. Nat. Microbiol. 2016, doi:10.1038/nmicrobiol.2016.55]. Can the author exclude that their results could be explained by heterogeneity in the chance to grow? Could they perhaps use their model to explore this eventuality? E.g. the model could have a random probability of single cells to restart growing.

2. Reply: *we thank the reviewer for this remark and suggestion. Indeed, we acknowledge that cells at the time of encapsulation may be 'compromised' or dead, since some 80% of cells stained propidium iodide positive in flow cytometry in the washed starting cell suspension (I. 170-172). We implicitly assumed that these individual dead cells will not be detected at later time points in the agarose beads by microscopy analysis, because their Syto9 signal would be below the detection limit. Furthermore, as we mentioned in figure 4d, this would result in 'no growth of both' or 'no growth of one' partner. However, we do agree with the reviewer that this may be insufficient and in particular in our models, we did not take into consideration that 80% of the cells of the starting community may be dead.*

Action: *We completely revised the modeling approach to include a proportion of dead cells in the starting mixture. We considered two scenarios: (i) cell death is randomly attributed to any cell, irrespective of the genotype, or (ii) cell death is occurring with a higher probability for fast-growing genotypes. In fact, we found that the second scenario explains much better the observed distribution of relative species abundances (explained in a new Figure 5b). We then re-analyzed (and report this in the revised version) all the previous interaction models, but including the 'proportional dead' scenario. This is described on I. 312-313, 342-345 and throughout the revised section I. 363-520, and in Supplementary methods, Section 2.3.*

As a second action point, we reanalyzed the experimental bead paired cell growth with a stricter cutoff to exclude potential dead cells (either for both or a single partner). The interactions were then compared by (i) scoring the proportions of observed growth pairs in a finer grid and comparing this to paired growth predictions from all models, and (ii) by calculating the ratio of all

paired growths (excluding assumed dead cells or dead pairs) and scoring the deviation of the ratio distribution curve to the null model (no interactions, no death), and to all other models. This deviation is then taken as the proportion of observed interactions that cannot be explained by differences in growth rates and would be classified 'positive' or 'negative'. This is shown in a new Figure 5h and 5i, and refines the statements of Fig. 4d.

My concern also applies to the statement at line 202: "In addition, only 19.7% (95% confidence interval 14.1–25.3) of beads with single occupancy surpassed a 10-fold productivity increase compared to the start, versus 47.4% (37.5–57.3, n = 12) for beads with two or more microcolonies. This indicated that being randomly partnered was on average beneficial for productivity.."

Is the result described here also consistent with the scenario where only one of the two randomly paired cells can grow? If only ~20% of species can grow in isolation up to 10 folds, then pairing them could lead to ~40% pairs growing to 10 folds, but where once of the two pairs doesn't grow at all.

3. Reply: *We appreciate the suggestion, but this analysis is independent of cell death. Beads with (observed) single occupancy are those, which either only had a single starter cell or a pair of which one was dead. If a starting cell is dead, it cannot contribute (much) to growth of the other partner. Beads with double occupancy are those where we recognized two microcolonies; hence, they must originate from two partners that were not dead at the start. The overall average positive effect of partnerships on paired growth in close distance is thus unchallenged by cell death. All models, except one, also predict significantly more than two-fold increase of paired growth compared to single bead occupancy. This is shown in a new Supplementary figure 4.*

Finally, the author could check if the fact that cells could randomly not grow would be able to explain the loss of diversity. In my belief, the authors discuss too quickly the role of non-growing cells; see line 383 : "Alternatively, since we worked with cells from natural communities, the estimated proportion of 20–30% non-dividing SC founder cells may have skewed community diversity upon regrowth."

4. Reply: *We appreciate this concern and the suggestion. It should be noted that modeling of multispecies communities remains very challenging and, as we mentioned, our efforts in this line can only be used conceptually to discern broad possible effects.*

Action: *To try and address this point, we have included in the revised model two scenarios; one in which cell death occurs randomly across all species, and the second, in which death is preferentially for rapidly growing species. We notice that random cell death has no effect on community growth and final diversity (Supplementary section 2.3.1), because community growth here is carbon limited, and a randomly assumed proportion of dead starting cells just leads to a smaller starting cell number, but not to different compositional diversity. In contrast, preferential cell death (indeed) has a stronger effect. Again, however, this is not sufficient to explain the dramatic loss of diversity. The diversity loss is best explained (conceptually) by a combination of factors: interspecific interactions that influence inherent growth rates plus a proportional death on fast-growers. We discuss this more in detail on l. 463-520; and in the revised figure 5 and a new figure 6.*

Overall my concern is that their experimental results can arise from alternative mechanisms than what the authors suggest and explore with their model. I think these alternative mechanisms w=should be discussed.

5. Reply: We very much appreciate this suggestion and we have done our best to model and discuss the effects of such mechanisms, notably, the presence of a large fraction of dead or growth-compromised cells from the beginning.

Action: see revised figure 5 and new figure 6, details on l. 367-374, l. 434-458, and new text l. 512-520; and in the discussion (l. 620-630).

****Minor comments****

75 “ the less likely that they may dominate the forming microcommunity of cells” unclear if “they” refers to species or to interactions.

6. Action: This referred to ‘interactions’. Was corrected to ‘such interactions’.

82 “a sort of egalitarianism at high species numbers” what do the authors mean with this? They first mention challenges, limitations and selection. What does it mean that these challenges, limitation and selections become egalitarian when there are multiple species? Can the author rephrase more specifically what they mean here?

7. Action: This referred to the previous l. 75; that the dominance of interspecific interactions on growth becomes less in a mixed system. Was corrected to ‘subdued interactions’.

94 I believe that the authors should comment here about the diffusibility across agar beads. The introduction suggests that “low connectivity” refers not just to no cell movement but also to no diffusion between microenvironments.

8. Reply: The reviewer rightfully assumes so, but with agarose beads we can only achieve restricted cell movement. We mentioned substrate diffusibility in l. 346.

Action: The following sentence was added “Embedding cells in agarose microbeads restrict cell movement but not substrate diffusion, although we expect that interspecific interactions mediated by e.g., diffusible molecules primarily affect paired species within individual beads.” (l. 106-108)

100 what do they mean with “deduce the scale and global nature of interspecific interactions”? I think here the formulation “scale and global nature” is not clear.

9. Reply and action: With ‘scale’ we meant ‘magnitude’ (e.g., strong or weak) – with ‘global nature’ we meant positive or negative; since we could not unravel individual types of interactions (e.g., typeVI secretion system or diffusible molecules). Phrase corrected as such.

106 Can the author elaborate or rephrase more directly the sentence “Despite expected increasing dominance of interspecific interactions, randomized partnerships were beneficial for community growth as opposed to growth in complete isolation.” Are the authors saying that, despite competitive interactions, randomize partnerships were beneficial compared to isolation?

10. Reply and action: The reviewer is correct. Our results show that on average (across the pools of measured beads) the productivity for paired species was higher than for species in single occupancy in individual beads. We apologize for the difficulty in terminology. This was rephrased to “were beneficial for growth of the community across all beads as opposed to growth

of cells inside single beads.” (L. 134-136) We also added a new Supplementary figure showing that in all model scenarios, except the *bimodal*, randomized partnerships have more than twofold more abundant growth than single isolated founder cells. L. 397-399

Do soil bacteria degrade agarose? I thought many did (e.g. see: Chi, Applied Microb. And Biotech, 2012, <https://link.springer.com/article/10.1007/s00253-012-4023-2>). Would consumption of agarose in one of the two environmental conditions (the low connectivity condition) affect the authors' main conclusions?

11. Reply and action: *Agarose (no matter how pure) contains diffusible sources of carbon, which a variety of bacteria can profit from. This is a general issue in single cell microscopy on e.g., agarose surfaces and is unavoidable to the usage of agarose. In contrast, we observed no detectable destruction of the agarose polymer and dissolution of beads as a result of agarase enzymatic activity. Given that we used carbon substrate mixtures to grow the communities (mixed C and soil extract), any further diffusible carbon from the agarose itself would have constituted an additional substrate, which one might have expected to favor growth in beads in comparison to liquid – which it didn't. Therefore, this is unlikely to have affected our conclusion (growth in beads diminishes diversity more than in liquid).*

Action: *no further action needed.*

232 it's not clear to me why higher differences would reflect faster and more abundant growth. The text says: “ higher magnitude of microcolony PBP differences than on mixed C substrates, reflecting the faster and more abundant growth on that substrate”

12. Reply: *The per-bead-productivity (or microcolony size) is a direct reflection of the amount of available carbon in the system, since the cells-in-beads grow in batch culture without replenishment of further substrates. Larger microcolonies therefore are most likely a consequence of more available substrate. When at the same time incubation point (e.g., diagrams at 24 and 48 h), individual microcolony sizes are larger on one substrate (soil extract) than another (mixed C) by at least a factor of 100 (= two logs), this must mean that those cells can grow faster. This is also apparent from the derived 'community growth rate' difference between soil extract and mixed C in Figure 2a.*

Action: *Was rephrased to: "showed 100-fold larger microcolony sizes than on mixed C substrates (e.g., Fig. 4c, 24 h)," (l. 275-277)*

352 the authors should talk about diffusible molecules also earlier, I believe.

13. Reply: *See remark above to l.94. Added to l. 105-108.*

587 “Overlapping signals were considered to originate from a Pve colony if the area overlap between two channels was greater than 30%.” Why 30%? That sounds like a low threshold to me.

14. Reply: *We appreciate the remark, but in contrast to the suggestion, 30% is a very conservative threshold setting. We assumed the scenario that when two microcolonies are very close together, the slight difference in brightness of the red and green fluorescence will cause overlapping areas. To be on the safe side, we assumed that when two areas (from red and green) overlapped at 30% or more, they would originate from a single microcolony.*

Action: *no further action needed.*

588 for the first time here “particle” is used to refer to a group of cells inside the bead. Before “particle” is present for various use. I’d suggest using another word other than particle. This was unclear to me.

15. Reply: *We apologize for the confusion. We changed this to ‘area’ or ‘colony’ throughout the text (it is in essence a pixel area that we detect, but in the slang of many image analysis programs, this is called ‘particle areas’).*

897 “the mean per bead productivity (PBP), or the product of the particle pixel area times its SYTO-9 fluorescence intensity normalized over all analyzed beads;” Two points: 1) I don’t understand why the productivity is calculated as an area x bead intensity. I assume it should be area x mean intensity. 2) how does the area increase related to the volume increase, which is the actual real measure of biomass increase? How do they go exactly from their area and intensity measure to a volumetric biomass estimation? Can they explain more clearly this calculation also in the supplementary text?

16. Reply: *We apologize. Productivity is indeed calculated as area times mean intensity. This was a typo.*

To the second question: since we cannot measure microcolonies in 3D, we had to use an approximation by including changes in pixel fluorescence intensity being the result of multiple cell layers. We assumed that when a microcolony increases in ‘height’, the fluorescence intensity per pixel increases as well as a result of multiple cell layers and the cell’s fluorescence shining through the layers above to some extent. This will not hold when there are too many cell layers, but for a few cell layers one can observe this. For this reason, we also used multiple exposure times and not used images where the signal intensity was saturated. As we explained in the revised supplementary text, this is an approximation, but the best that we could think of. Obviously, since cells from different species can also have different intensities from Syto9 staining, we cannot truly estimate biovolumes from area times mean intensity, but it is an approximation. We don’t use biovolume estimates at all in the results, only the per-bead-productivity in units of area x mean pixel intensity (see the axis scales in all figures).

Action: *We rephrased and detailed this in a revised Supplementary text section.*

665 typo “slowed” would be “slower”. English should be revised here.

17. Reply: *The model section was completely revised in the new model inclusion.*

680 “and 60% chance for a factor in between 0.8–1.4 that could increase the growth rate.” How can the factor 0.8 increase the growth rate? Did I miss something?

18. Reply and action: *The reviewer is correct that not all random produced factors will increase (the sentence said: “a random factor that **could** increase growth rate”.) However, to avoid ambiguity, we changed this to “..factor that can moderate or increase growth rate”. (I. 930)*

940 and other occurrence in main text and SI: can the authors write log₁₀ PBP more precisely? Where 10 is clearly the base of the logarithm?

19. Reply: *We followed MatLab style that ‘log₁₀’ means the 10 base of the log. We will leave this for the journal to indicate the appropriate style ¹⁰log or log₁₀. In the figures we changed this to log₁₀.*

Fig 5b mu seems to be not defined

Fig 5d there is no reference in text to this panel.

Fig 5e there is a typo on the numbers on x axis

Fig 5f I find the heat map (right) hard to read. For example, what should the reader see about the correlations between models? I'd consider making a simpler figure, that conveys the message the authors wanted to convey with this more complicated figure. Note also that Fig 5f (left plot) has axis with no units.

20. Reply and action: We completely revised this figure with the new models (all panels are new) and have tried to take into account the above remarks.

New figure 6. Panels a, b and d were calculated with the new models but are similar to the previous fig. 5e. Panels c and e are new data representations.

Reviewer #2 (Remarks to the Author):

In this manuscript, the authors explore and investigate the role of environmental connectivity in shaping community growth and species diversity and underlying interspecific soil microbial interactions. In particular, through laboratory experiments, they studied the growth of naturally-derived soil microbial communities and demonstrate that low environmental connectivity appears to reduce global community diversity as compared to high connectivity conditions. In addition, via computational community growth modeling, they further demonstrate that this microbial diversity loss seems to result from biased negative microbial interactions. Their results lend support to the importance of interspecific interactions in shaping natural microbial communities, which need to be considered and fundamentally understood to move towards their control in medicine and biotechnologies.

This work combining both experiments and modeling to investigate the influence of environmental connectivity on microbial community growth and diversity is relevant, well designed, and novel. Overall, their analyses are pertinent and sound, the results are well presented, and the manuscript is well written. Overall, I have only minor comments, which are

detailed below. Thus, I think this manuscript should be considered for publication in Communications Biology, once the authors have successfully addressed the recommended revisions.

21. Reply: *We thank the reviewer for the summary and for the positive criticisms on our work. We have done our best to accommodate the reviewer's suggestions as best as possible.*

Minor comments:

- The authors studied the effect of environmental connectivity on the growth and diversity of soil microbial communities. Although their results may potentially apply to microbial communities from other biomes (e.g. host-associated, marine), it is not demonstrated in the present study. Thus, I would recommend, where appropriated (for example in the title), to specify that the results and conclusions presented in the manuscript apply only to soil microbial communities.

22. Reply: *This is correct. We suggested that this effect may have consequences for a wider range of environments and biomes, but only have shown it for soil microbial communities in agarose beads. We changed the title as suggested to “**Environmental Connectivity Controls Diversity in Soil Microbial Communities**” and further throughout the text (e.g., l. 47). However, we maintain that it is worth to investigate whether this could be something more general (or different) to other microbiomes (discussion lines l. 629-632).*

- Different methodologies were used to quantify biomass in low versus high connectivity conditions. Although comparing SC yields as the ratio to *P. veronii* is appropriate, it is biased by the actual growth ability of this specific species in both conditions. Thus, a direct comparison of relative biomasses through time in both conditions may be complementary to their approach.

23. Reply: *The reviewer is correct that we used different methodologies to measure community biomass in liquid (flow cytometry cell counting) and beads (imaging). Unfortunately, embedding cells in agarose beads makes it impossible to release them during or after the experiment (except for isolating their DNA). We compared growth and growth rates as a function of time (Fig. 2a and b), which were very similar, and a comparison of their maximum yield in Fig. 2d indicated that there is no significant difference in the ratio.*

In addition, we remeasured the DNA concentrations from those samples, which were in the same order of magnitude – suggesting that the biomass growth was similar in both systems. Therefore, we feel confident that this conclusion is correctly drawn from the data.

Action: *We added the DNA concentrations as a Supplementary table (l. 196).*

- Results presented in Fig.3a may benefit from a more comprehensive representation of all experimental results (through time) by comparing beta-diversity across all samples, for example via a Principal Coordinates Analysis. This would allow visualizing the evolution of the global community compositions, and how they relate to each other along the course of the experiments.

24. Reply and action: *We appreciate the suggestion and have added beta-diversity comparison using Phyloseq on normalized OTU relative abundances of individual replicates as a **new figure 3c panel**. Results description to this is presented in l. 219-221.*

- In the discussion, the authors state the hypothesis “that low connectivity leads to diversity loss because low connectivity environments can contain fewer starting cells, . . . , in comparison to high connectivity and higher starting species numbers”. Here, the authors assume that low connected environments may contain fewer cells, but I must say I am not sure to understand the rationale behind this assumption. Since (soil) microbial communities are highly dynamic, a low connected environment may also contain a high diversity. Thus, I suggest the authors clarify and elaborate on this hypothesis.

25. Reply: We thank the reviewer for the remark and apologize for the confusion here. We argue that low connectivity environments are those that as a result of their physical nature can only contain few cells; and, therefore, only few different genotypes. One would expect this to occur in, e.g., small soil pores – as illustrated in Figure 1. The ‘soil’ would then be an aggregate of many low connectivity environments, and a soil microbial community should in this assumption be seen as a ‘meta-community’ of individual environments, with each individual environment being dependent on the smaller number of species temporally being trapped and isolated. As we discuss in I. 535-539 and I. 590-611, since (aggregate) soil microbial communities are highly diverse, one would have to assume that the many isolated environments can become dynamically and temporally connected.

Action: We explained this more precisely in the introduction in lines **88-91** and in the legend to Figure 1 (l. 1174-1177.)

REVIEWERS' COMMENTS:

Reviewer #1 (Remarks to the Author):

The authors have addressed all my major comments. I have still a number of minor comments. These minor comments all point to the need to revise the text before publication. Overall, I think that the paper is improved. However the new text is sometimes hard to read and needs a careful revision.

Abstract: are the terms "biased positive interspecific interactions" and "biased negative interspecific interactions" clear enough for an abstract? I think that it is unclear what biased positive and biased negative mean without reading the text. I'd use e.g. mostly negative and mostly positive.

Line 88: In the current formulation of this sentence it is unclear why "small isolated spaces" have stronger species-species interactions compared to "larger connected living environments". The focus of this new sentence seems to be on the size of the environment, while I guess that what matters is whether cells interact with a many or few other cells. If the authors should here want to talk about the level of connectivity of the environment, then maybe they should rephrase as now they seem to make a point on how large the environment is.

Here the new sentence:

The connectivity of any environment is thus expected to impose a range of challenges, limitations and selections to microbial growth. Small isolated spaces can only hold a few cells and genotypes, leading to isolated growth but dominating interspecific interactions. The larger is the connected living environment, the more genotypes are likely to be found and sustained, but the effect of interspecific interactions is subdued.

[This sentence replaced: The connectivity of any environment is expected to impose a range of challenges, limitations and selections to microbial growth, which translate in a gradient from isolated growth and survival, to dominating interspecific interactions at low species numbers, and potentially, to a sort of egalitarianism at high species numbers.]

Line 108:

This sentence could be more clear:

"Embedding cells in agarose microbeads restricts cell movement but not substrate diffusion, although we expect that interspecific interactions mediated by e.g., diffusible molecules primarily affect paired species at short distances within and not between individual beads".

The authors say that embedding cells in agarose does not restrict substrate diffusion (within the bead -I think they mean). But they say that, despite substrate diffusion not being restricted, species don't affect each other across different beads. I think the latter is a separate statement.

Line 132: I'd revise punctuation and add "environments": Our results, supported by community growth simulations, indicate that a pool of fragmented homogenously sized low connectivity environments disfavors maintenance of microbial diversity compared to high connectivity **environments**.

223: here it's "diverse" or "different"?

288 "The majority of paired growth ratios is more than tenfold deviating". Deviating from what?

388: Is "affirming" is the right word? "Confirming" maybe?

367: I find this sentence quite hard to follow. I'd try to make it more reader friendly: "The resulting simulated microcolony size distributions at stationary phase across all beads with single occupancy were skewed to smaller sizes in the case of growth penalty and a proportionally higher death at start for fast-growers, and to larger sizes in absence of either (Fig. 5b), or with randomly attributed cell death at start (Supplementary methods, Section 2.3). Of these, the smaller size skewed distribution of proportional death and single growth penalty was the most similar to experimental observations of microcolony PBPs (Fig. 5c).

373 Revise English: This supported the assumption that single occupancy growth under low connectivity conditions is indeed penalized, and suggested further that a significant fraction of particularly the fast-growing taxa are compromised for growth when recovered from the soil.

436 What does invoked mean? "invoked interspecific interactions"

450 " in simulations and from experimental data" use "in" for both.

459: Revise the expression "interaction... are non-random mixture of positive and negative signs". What does it mean that interactions are signs? Here the full sentence: "but is further subject to interspecific interactions that are a non-random mixture of positive ('beneficial') and negative ('competitive') signs as in the biased positive model."

513 Revise sentence ("to within" seems wrong): "Simulations explained on average 38.0% and 51.3% of individual OTU abundances to within a range of two- and fourfold, respectively"

517: Revise punctuation and English: "This showed that, although obviously, we cannot predict the individual taxa behaviour very precisely, broad-scale modeling of growth rate distributions and interspecies interactions captured relevant trends of community behaviour."

630: I think here the Authors should use a softer expression for the results of their simulations, such as "simulations suggest" rather than: "Further surprising outcome of the simulations was that fast-growing and abundant genotypes have a higher probability to being dead in natural (freshly extracted) sand communities than slow-growing less abundant ones.